# SAMPLE-EFFICIENT TRAINING FOR DIFFUSION

## ABSTRACT

Score-based diffusion models have become the most popular approach to deep generative modeling of images, largely due to their empirical performance and reliability. Recently, a number of theoretical works (Chen et al.; 2022a; 2023; Benton et al., 2023) have shown that diffusion models can efficiently sample, assuming $L^2$-accurate score estimates. The score-matching objective naturally approximates the true score in $L^2$, but the sample complexity of existing bounds depends *polynomially* on the data radius and desired Wasserstein accuracy. By contrast, the time complexity of sampling is only logarithmic in these parameters. We show that estimating the score in $L^2$ *requires* this polynomial dependence, but that a number of samples that scales polylogarithmically in the Wasserstein accuracy actually do suffice for sampling. We show that with a polylogarithmic number of samples, the ERM of the score-matching objective is $L^2$ accurate on all but a probability $\delta$ fraction of the true distribution, and that this weaker guarantee is sufficient for efficient sampling.

## 1 INTRODUCTION

Score-based diffusion models are currently the most successful methods for image generation, serving as the backbone for popular text-to-image models such as stable diffusion (Rombach et al., 2022), Midjourney, and DALL·E 2 (Ramesh et al., 2022) as well as achieving state-of-the-art performance on other audio and image generation tasks (Sohl-Dickstein et al., 2015; Ho et al., 2020; Jalal et al., 2021; Song et al., 2022; Dhariwal & Nichol, 2021).

The goal of score-based diffusion is to sample from a (potentially complicated) distribution $q_0$. This involves two components: *training estimates* of score functions from samples, and *sampling* using the trained estimates. To this end, consider the following stochastic differential equation, which is often referred to as the *forward* SDE:

$$\mathrm{d}x_t = -x_t \, \mathrm{d}t + \sqrt{2} \, \mathrm{d}B_t, \quad x_0 \sim q_0 \tag{1}$$

where $B_t$ represents Brownian motion. Here, $x_0$ is a sample from the original distribution $q_0$ over $\mathbb{R}^d$, while the distribution of $x_t$ can be computed to be

$$x_t \sim e^{-t}x_0 + \mathcal{N}(0, \sigma_t^2 I_d)$$

for $\sigma_t^2 = 1 - e^{-2t}$. Note that this distribution approaches $\mathcal{N}(0, I_d)$, the stationary distribution of (1), exponentially fast.

Let $q_t$ be the distribution of $x_t$, and let $s_t(y) := \nabla \log q_t(y)$ be the associated *score* function. We refer to $q_t$ as the $\sigma_t$-*smoothed* version of $q_0$. Then, starting from a sample $x_T \sim q_T$, there is a reverse SDE associated with the above forward SDE in equation 1 (Anderson, 1982):

$$\mathrm{d}x_{T-t} = (x_{T-t} + 2s_{T-t}(x_{T-t})) \, \mathrm{d}t + \sqrt{2} \, \mathrm{d}B_t. \tag{2}$$

That is to say, if we begin at a sample $x_T \sim q_T$, following the reverse SDE in equation 2 back to time $0$ will give us a sample from the *original* distribution $q_0$. This suggests a natural strategy to sample from $q_0$: start at a time $T$ large enough using a sample from $\mathcal{N}(0, I_d)$, and follow the reverse SDE back to time $0$. Since $x_T$ is distributed exponentially close in $T$ to $\mathcal{N}(0, I_d)$, our samples at time $0$ will end up being distributed close to $q_0$. In particular, if $T$ is large enough–logarithmic in $\frac{m_2}{\varepsilon}$–then our samples from this process will be $\varepsilon$-close in TV to being drawn from $q_0$. Here $m_2^2$ is the second moment of $q_0$, given by

$$m_2^2 := \mathop{\mathbb{E}}_{x \sim q_0} \left[ \|x\|^2 \right].$$

Of course, to follow the reverse SDE from equation 2, we need access to the *score* functions $s_t$ at various times $t$. A diffusion model *estimates* these score functions using *score-matching*.

**Score-matching.** One can show (Theorem 1, Hyvärinen (2005)) that the score function at time $t$, $s_t$, is the minimizer over functions $s$ of the following "score matching objective":

$$\mathbb{E}_{x \sim q_0, z \sim \mathcal{N}(0, \sigma_t^2 I_d)} \left[ \left\| s(e^{-t}x + z) - \frac{-z}{\sigma_t^2} \right\|_2^2 \right] \tag{3}$$

This gives a natural method for estimating $s_t$: minimize the *empirical* value of equation 3 over samples $x_i \sim q_0$ and independent Gaussian samples $z_i \sim \mathcal{N}(0, \sigma_t^2 I_d)$. In other words, for a given function family $\mathcal{H}$ of candidate score functions, estimate the empirical risk minimizer (ERM):

$$\underset{\widehat{s}_t \in \mathcal{H}}{\arg\min} \quad \frac{1}{m} \sum_{i=1}^{m} \left\| \widehat{s}_t(e^{-t}x_i + z_i) - \frac{-z_i}{\sigma_t^2} \right\|_2^2 \tag{4}$$

Such a minimizer can be approximated in practice using deep neural networks, with the objective optimized by stochastic gradient descent (SGD).

**Sampling.** Given the score estimates $\widehat{s}_t$, one can sample from a distribution close to $q_0$ by running the reverse SDE in equation 2.

To practically implement the reverse SDE in equation 2, we discretize this process into $N$ steps and choose a sequence of times $0 = t_0 < t_1 < \cdots < t_N < T$. At each discretization time $t_k$, we use our score estimates $\widehat{s}_{t_k}$ and proceed with an *approximate* reverse SDE using our score estimates, given by the following. For $t \in [t_k, t_{k+1}]$,

$$\mathrm{d}x_{T-t} = (x_{T-t} + 2\widehat{s}_{T-t_k}(x_{T-t_k}))\,\mathrm{d}t + \sqrt{2}\,\mathrm{d}B_t. \tag{5}$$

Here, we will begin at $x_T = x_{T-t_0} \sim \mathcal{N}(0, I_d)$. We will let $\widehat{q}_t$ be the distribution at time $t$ from following the above SDE. This algorithm is referred to as "DDPM", as defined in (Ho et al., 2020).

A number of recent theoretical works (Chen et al.; 2022a; 2023; Benton et al., 2023) have studied this sampling process: how many steps are needed, what is the right schedule of discretization times $t$, and how accurately must the score be estimated? They have shown polynomial time algorithms in remarkable generality.

For example, consider *any* $d$-dimensional distribution $q_0$ supported in $B(0, R)$ (or, more generally, subgaussian with parameter $R^2$). In Chen et al. it was shown that the SDE can sample from a distribution $\varepsilon$-close in TV to a distribution $(\gamma \cdot R)$-close in 2-Wasserstein to $q_0$, in $\text{poly}(d, \frac{1}{\varepsilon}, \frac{1}{\gamma})$ steps, as long as the score estimates are close in $L^2$. That is, as long as:

$$\mathbb{E}_{x \sim q_t} \left[ \|s_t(x) - \widehat{s}_t(x)\|^2 \right] \leq O^*\left(\varepsilon^2\right) \tag{6}$$

where $O^*(\cdot)$ hides factors logarithmic in $d$, $\frac{1}{\varepsilon}$, and $\frac{1}{\gamma}$.

Moreover, in the same situation, Block et al. (2020) showed that *training* with the score-matching objective achieves the desired $L^2$ accuracy with a similar sample complexity to the above time complexity. The precise bound depends on the hypothesis class $\mathcal{H}$; for finite hypothesis classes, it is $\text{poly}(d, \varepsilon, \frac{1}{\gamma}, \log |\mathcal{H}|)$.

On the *sampling* side, more recent work (Chen et al., 2022a; Benton et al., 2023) has given exponentially better dependence on the accuracy $\gamma$, as well as replacing the uniform bound $R$ by the second moment. In particular, Benton et al. (2023) show that $\widetilde{O}(\frac{d}{\varepsilon^2} \log^2 \frac{1}{\gamma})$ steps suffice to sample from a distribution that is $\gamma \cdot m_2$ close to $q_0$ in 2-Wasserstein, as long as the score estimates $\widehat{s}_t$ at these times $t$ are accurate enough – that is, satisfy:

$$\mathbb{E}_{x \sim q_t} \left[ \|s_t(x) - \widehat{s}_t(x)\|^2 \right] \leq O^*\left(\frac{\varepsilon^2}{\sigma_t^2}\right). \tag{7}$$

This requirement is easier to satisfy than that of equation 6, because the schedule is such that $\sigma_t \leq 1$ always.

Thus, for *sampling*, recent work has replaced the $\text{poly}(\frac{1}{\gamma})$ dependence with $\text{poly}(\log \frac{1}{\gamma})$ to sample from a distribution $\gamma \cdot m_2$ close to $q_0$ in 2-Wasserstein. Can we do the same for the sample complexity of *training*? That is the question we address in this work.

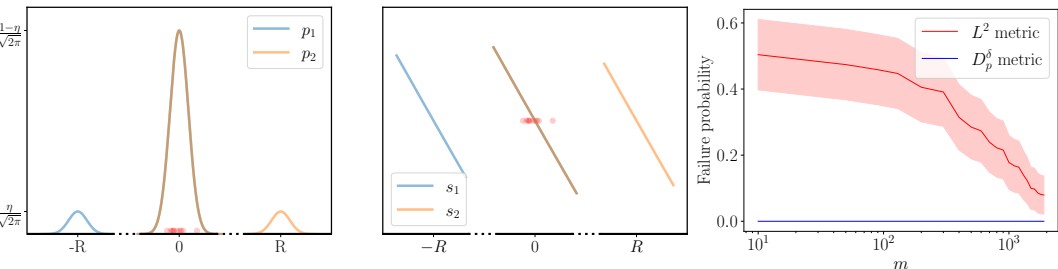

Figure 1: Given $o\left(\frac{1}{\eta}\right)$ samples from either $p_1 = (1-\eta)\mathcal{N}(0, 1) + \eta\mathcal{N}(-R, 1)$, or $p_2 = (1-\eta)\mathcal{N}(0, 1) + \eta\mathcal{N}(R, 1)$ we will only see samples from the main Gaussian with high probability, and cannot distinguish between them. However, if we pick the wrong score function, the $L^2$ error incurred is large - about $\eta R^2$. On the right, we take $\eta = 0.001, R = 10000, \delta = 0.01$. We plot the probability that the ERM has error larger than 0 in the $L^2$ sense, and our $D_p^\delta$ sense.

**Our results.** Ideally, one would like to simply show that the ERM of the score matching objective will have bounded $L^2$ error (7) with a number of samples that scales polylogarithmically in $\frac{1}{\gamma}$. Unfortunately, this is *false*. In fact, it is information-theoretically impossible to achieve equation 7 in general without $\text{poly}(\frac{1}{\gamma})$ samples. See Figure 1, or the discussion in Section 4, for a hard instance.

In the example in Figure 1, score matching + DDPM still *works* to sample from the distribution with sample complexity scaling with $\text{poly}(\log \frac{1}{\gamma})$; the problem lies in the theoretical justification for it. Given that it is impossible to learn the score in $L^2$ to sufficient accuracy with fewer than $\text{poly}(\frac{1}{\gamma})$ samples, such a justification needs a different measure of estimation error. We will introduce such a measure, showing (1) that it will be small for all relevant times $t$ after a number of samples that scales *polylogarithmically* in $\frac{1}{\gamma}$, and (2) that this measure suffices for fast sampling via the reverse SDE.

The problem with measuring error in $L^2$ comes from outliers: rare, large errors can increase the $L^2$ error while not being observed on the training set. We propose a more robust measure of distance, the $1 - \delta$ quantile error. For distribution $p$, and functions $f, g$, we say that

$$D_p^\delta(f, g) \leq \varepsilon \iff \Pr_{x \sim p}\left[\|f(x) - g(x)\|_2 \geq \varepsilon\right] \leq \delta. \tag{8}$$

Our main result shows that for a finite-size function class $\mathcal{H}$ containing a sufficiently accurate score estimate, we can estimate the score function at a given time $t$ in our robust distance measure using a number of samples that is independent of the domain size or the maximum value of the score.

**Theorem 1.1** (Score Estimation for Finite Function Class). *For any distribution $q_0$ and time $t > 0$, consider the $\sigma_t$-smoothed version $q_t$ with associated score $s_t$. For any finite set $\mathcal{H}$ of candidate score functions. If there exists some $s^* \in \mathcal{H}$ such that*

$$\mathbb{E}_{x \sim q_t}\left[\|s^*(x) - s_t(x)\|_2^2\right] \leq \frac{\delta_{score} \cdot \delta_{train} \cdot \varepsilon^2}{100 \cdot \sigma_t^2}, \tag{9}$$

*for a sufficiently large constant $C$, then using $m > \widetilde{O}\left(\frac{1}{\varepsilon^2 \delta_{score}}(d + \log \frac{1}{\delta_{train}})\log \frac{|\mathcal{H}|}{\delta_{train}}\right)$ samples, the empirical minimizer $\hat{s}$ of the score matching objective as described in equation 4 used to estimate $s_t$ satisfies*

$$D_{q_t}^{\delta_{score}}(\hat{s}, s_t) \leq \varepsilon/\sigma_t$$

*with probability $1 - \delta_{train}$.*

Theorem 1.1 supposes that $\mathcal{H}$ is a finite hypothesis class. One can extend this to infinite classes by taking a union bound over a net; see Theorem 1.5 for an application to neural networks, getting a bound polynomial in the number of parameters and logarithmic in the maximum weight and $m_2$.

Note that for distributions supported in $B(0, R)$, the final step of the sampling process has $\sigma_t = \gamma$, for which the score can be as large as $\frac{R}{\gamma^2}$. Prior work (Block et al., 2020) requires $\text{poly}(\frac{R}{\gamma})$ samples to learn the score in $L^2$ in this setting. In contrast, the sample complexity in Theorem 1.1 is independent of the domain size and smoothing level.

Previous results like (Benton et al., 2023; Chen et al.) show that learning a score estimate in $L^2$ suffices for efficient sampling via DDPM (described in equation 5). We adapt (Benton et al., 2023) to show that learning the score in our new outlier-robust sense also suffices for sampling, if we have accurate score estimates at each relevant discretization time.

**Theorem 1.2.** *Let $q$ be a distribution over $\mathbb{R}^d$ with second moment $m_2^2$ between $1/poly(d)$ and $poly(d)$. For any $\gamma > 0$, there exist $N = \widetilde{O}(\frac{d}{\varepsilon^2 + \delta^2} \log^2 \frac{1}{\gamma})$ discretization times $0 = t_0 < t_1 < \cdots < t_N < T$ such that if the following holds for every $k \in \{0, \ldots, N-1\}$:*

$$D_{q_{T-t_k}}^{\delta/N}(\widehat{s}_{T-t_k}, s_{T-t_k}) \leq \frac{\varepsilon}{\sigma_{T-t_k}}$$

*then the SDE process in equation 5 can produce a sample from a distribution that is within $\widetilde{O}(\delta + \varepsilon\sqrt{\log(d/\gamma)})$ in $\mathsf{TV}$ distance to $q_\gamma$ in $N$ steps.*

Applying Theorem 1.1 with a union bound, and combining with Theorem 1.2, we have the end-to-end guarantee:

**Corollary 1.3** (End-to-end Guarantee)**.** *Let $q$ be a distribution over $\mathbb{R}^d$ with second moment $m_2^2$ between $1/poly(d)$ and $poly(d)$. For any $\gamma > 0$, there exist $N = \widetilde{O}(\frac{d}{\varepsilon^2 + \delta^2} \log^2 \frac{1}{\gamma})$ discretization times $0 = t_0 < \cdots < t_N < T$ such that if $\mathcal{H}$ contain approximations $h_{T-t_k}$ that satisfy*

$$\mathbb{E}_{x \sim q_t} \left[ \|h_{T-t_k}(x) - s_{T-t_k}(x)\|^2 \right] \leq \frac{\delta \cdot \varepsilon^3}{CN^2 \sigma_{T-t_k}^2} \cdot \frac{1}{\log \frac{d}{\gamma}}$$

*for sufficiently large constant $C$, then given $m = \widetilde{O}\left( \frac{N}{\varepsilon^3}(d + \log \frac{1}{\delta}) \log \frac{|\mathcal{H}|}{\delta} \log \frac{1}{\gamma} \right)$ samples, with $1 - \delta$ probability the SDE process in equation 5 can sample from a distribution $\varepsilon$-close in $\mathsf{TV}$ to a distribution $\gamma m_2$-close in 2-Wasserstein to $q$ in $N$ steps.*

The sample complexity required for the end-to-end guarantee above depends polynomially on $N$, the number of discretization times required for sampling. As stated, this depends polylogarithmically on $\frac{1}{\gamma}$, for a final sample complexity that scales polylogarithmically in $\frac{1}{\gamma}$. Prior work (Block et al., 2020) needs $\text{poly}(\frac{1}{\gamma})$ training samples even to learn the score at the final smoothing level, as described earlier. Thus our bound gives much higher Wasserstein accuracy for a fixed number of samples.

**Remark 1.4** (Scaling of $m_2$)**.** *The above theorems assume $m_2^2$ is between $\frac{1}{poly(d)}$ and $poly(d)$ because the bound is simpler to state with this assumption, and because this is the "right" scaling. The forward diffusion process transforms any arbitrary distribution into a standard normal distribution of second moment $d$. If the initial moment $m_2^2$ is extremely large or extremely small relative to $d$, the convergence time will have an additional (logarithmic) factor depending on $m_2$. This dependence is analyzed explicitly in Theorem B.6 and Corollary C.1. However, a better way to handle these situations is to rescale the distribution to have second moment polynomial in $d$, so that Theorem 1.2 and Corollary 1.3 hold for the rescaled distributions.*

**Application to Neural Networks.** To demonstrate that the restriction of Theorem 1.1 to finite hypothesis classes is relatively mild, we show that it implies results for general neural networks with ReLU activation. If the score function is well-approximated by a depth-$D$, $P$-parameter neural network, then roughly $dPD$ samples suffice to learn it well enough for accurate sampling.

**Theorem 1.5** (Score Training for Neural Networks)**.** *For any distribution $q_0$ with second moment $m_2^2$, and any time $t > 0$, let $q_t$ be the $\sigma_t$-smoothed version with associated score $s_t$. Let $\phi_\theta(\cdot)$ be the*

*fully connected neural network with ReLU activations parameterized by $\theta$, with $P$ total parameters and depth $D$. If there exists some weight vector $\theta^*$ with $\|\theta^*\|_F \leq \Theta$ such that*

$$\mathop{\mathbb{E}}_{x \sim q_t} \left[ \|\phi_{\theta^*}(x) - s_t(x)\|^2 \right] \leq \frac{\delta_{score} \cdot \delta_{train} \cdot \varepsilon^2}{1000 \cdot \sigma_t^2}$$

*then using $m > \widetilde{O}\left( \frac{(d + \log \frac{1}{\delta_{train}}) \cdot PD}{\varepsilon^2 \delta_{score}} \cdot \log\left( \frac{(m_2 + \sigma)\Theta}{\delta_{train}} \right) \right)$ samples, the empirical minimizer $\phi_{\widehat{\theta}}$ of the score matching objective used to estimate $s_t$ (over $\phi_\theta$ with $\|\theta\|_F \leq \Theta$) satisfies*

$$D_{q_t}^{\delta_{score}}(\phi_{\widehat{\theta}}, s_t) \leq \varepsilon/\sigma.$$

*with probability $1 - \delta_{train}$.*

## 2 RELATED WORK

Score-based diffusion models were first introduced in (Sohl-Dickstein et al., 2015) as a way to tractably sample from complex distributions using deep learning. Since then, many empirically validated techniques have been developed to improve the sample quality and performance of diffusion models (Ho et al., 2020; Nichol & Dhariwal, 2021; Song & Ermon, 2020; Song et al., 2021b;a). More recently, diffusion models have found several exciting applications, including medical imaging and compressed sensing (Jalal et al., 2021; Song et al., 2022), and text-to-image models like DALL·E 2 (Ramesh et al., 2022) and Stable Diffusion (Rombach et al., 2022).

Recently, a number of works have begun to develop a theoretical understanding of diffusion. Different aspects have been studied – the sample complexity of training with the score-matching objective (Block et al., 2020), the number of steps needed to sample given accurate scores (Chen et al.; 2022a;b; 2023; Benton et al., 2023; Bortoli et al., 2021; Lee et al., 2023), and the relationship to more traditional methods such as maximum likelihood (Pabbaraju et al., 2023; Koehler et al., 2023).

On the *training* side, Block et al. (2020) showed that for distributions bounded by $R$, the score-matching objective learns the score of $q_\gamma$ in $L^2$ using a number of samples that scales polynomially in $\frac{1}{\gamma}$. On the other hand, for *sampling* using the reverse SDE in equation 2, (Chen et al., 2022a; Benton et al., 2023) showed that the number of steps to sample from $q_\gamma$ scales polylogarithmically in $\frac{1}{\gamma}$ given $L^2$ approximations to the scores.

Our main contribution is to show that while learning the score in $L^2$ *requires* a number of samples that scales polynomially in $\frac{1}{\gamma}$, the score-matching objective *does learn* the score in a weaker sense with sample complexity depending only *polylogarithmically* in $\frac{1}{\gamma}$. Moreover, this weaker guarantee is sufficient to maintain the polylogarithmic dependence on $\frac{1}{\gamma}$ on the number of steps to sample with $\gamma \cdot m_2$ 2-Wasserstein error.

## 3 PROOF OVERVIEW

### 3.1 TRAINING

We show that the score-matching objective (equation 4) concentrates well enough that the ERM is close to the true minimizer.

**Background: the true expectation gives the true score.** For a fixed $t$, let $\sigma = \sigma_t$ and $p$ be the distribution of $e^{-t}x$ for $x \sim q_0$. We can think of a joint distribution of $(y, x, z)$ where $y \sim p$ and $z \sim N(0, \sigma^2 I_d)$ are independent, and $x = y + z$ is drawn according to $q_t$. The score matching objective is then

$$\mathop{\mathbb{E}}_{x,z} \left[ \left\| s(x) - \frac{-z}{\sigma^2} \right\|_2^2 \right]$$

Because $x = y + z$ for Gaussian $z$, Tweedie's formula states that the true score $s^*$ is given by

$$s^*(x) = \mathop{\mathbb{E}}_{z|x} \left[ \frac{-z}{\sigma^2} \right].$$

Define $\Delta = s^*(x) - \frac{-z}{\sigma^2}$, so $\mathbb{E}[\Delta \mid x] = 0$. Therefore for any $x$,

$$l(s, x, z) := \left\| s(x) - \frac{-z}{\sigma^2} \right\|_2^2 = \|s(x) - s^*(x) + \Delta\|^2$$

$$= \|s(x) - s^*(x)\|^2 + 2\langle s(x) - s^*(x), \Delta \rangle + \|\Delta\|^2. \qquad (10)$$

For every $x$, the second term is zero on average over $(z \mid x)$, so the expected loss is

$$\mathbb{E}_{x,z}[l(s, x, z)] = \mathbb{E}_{x,z}[\|s(x) - s^*(x)\|^2 + 2\langle s(x) - s^*(x), \Delta \rangle + \|\Delta\|^2]$$

$$= \mathbb{E}_x[\|s(x) - s^*(x)\|^2] + \mathbb{E}_{x,z}[\|\Delta\|^2]$$

The $\|\Delta\|^2$ term is independent of $s$, so indeed the score matching objective is minimized by the true score. Moreover, an $\varepsilon$-approximate optimizer of $l(s)$ will be close in $L^2$, as needed by prior samplers.

**Understanding the ERM.** The algorithm chooses the score function $s$ minimizing the empirical loss,

$$\hat{\mathbb{E}}_{x,z}[l(s, x, z)] := \frac{1}{m} \sum_{i=1}^m l(s, x_i, z_i) = \hat{\mathbb{E}}_{x,z}[\|s(x) - s^*(x)\|^2 + 2\langle s(x) - s^*(x), \Delta \rangle + \|\Delta\|^2].$$

Again the $\hat{\mathbb{E}}[\|\Delta\|^2]$ term is independent of $s$, so it has no effect on the minimizer and we can drop it from the loss function. Let

$$l'(s, x, z) := \|s(x) - s^*(x)\|^2 + 2\langle s(x) - s^*(x), \Delta \rangle \qquad (11)$$

so $l'(s^*, x, z) = 0$ always, $\mathbb{E}[l'(s, x, z)] = \mathbb{E}[\|s(x) - s^*(x)\|^2]$, and we want to show that

$$\hat{\mathbb{E}}_{x,z}[l'(s, x, z)] > 0 \qquad (12)$$

for all candidate score functions $s$ that are "far" from $s^*$. We will show equation 12 is true with high probability for each individual $s$, then take a union bound (possibly over a net).

**Boundedness of $\Delta$.** Now, $z \sim N(0, \sigma^2 I_d)$ is technically unbounded, but is exponentially close to being bounded: $\|z\| \lesssim \sigma\sqrt{d}$ with overwhelming probability. You are certainly unlikely to sample any $z_i$ much larger than this. So for the purpose of this proof overview, imagine that $z$ were drawn from a distribution of bounded norm, i.e., $\|z\| \le B\sigma$ always; the full proof needs some exponentially small error terms to handle the tiny mass the Gaussian places outside this ball. Then since $\Delta = \frac{z}{\sigma^2} - \mathbb{E}_{z|x}[\frac{z}{\sigma^2}]$, $\|\Delta\| \le 2B/\sigma$ as well.

**Warmup: poly$(R/\sigma)$.** As a warmup, consider the setting of prior work Block et al. (2020): (1) $\|x\| \le R$ always, so $\|s^*(x)\| \lesssim \frac{R}{\sigma^2}$; and (2) we only optimize over candidate score functions $s$ with $\|x\| \lesssim \frac{R}{\sigma^2}$, so $\|s(x) - s^*(x)\| \lesssim \frac{R}{\sigma^2}$. With both these restrictions, then, $|l'(s, x, z)| \le \frac{R^2}{\sigma^4} + \frac{RB}{\sigma^3}$. We can then apply a Chernoff bound to show concentration of $l'$: for poly$(\varepsilon, \frac{R}{\sigma}, B, \log\frac{1}{\delta_{\text{train}}})$ samples, with $1 - \delta_{\text{train}}$ probability we have

$$\hat{\mathbb{E}}_{x,z}[l'(s, x, z)] \ge \mathbb{E}_{x,z}[l'(s, x, z)] - \frac{\varepsilon}{\sigma^2} = \mathbb{E}[\|s(x) - s^*(x)\|^2] - \frac{\varepsilon^2}{\sigma^2}$$

which is greater than zero if $\mathbb{E}[\|s(x) - s^*(x)\|^2] > \frac{\varepsilon^2}{\sigma^2}$. Thus the ERM would reject each score function that is far in $L^2$. However, as we show in Section 4, both these restrictions are necessary: the score matching ERM needs a polynomial dependence on both the distribution norm and the candidate score function values to learn in $L^2$. To avoid these, we settle for rejecting score functions $s$ that are far in our stronger distance measure $D_p^{\delta_{\text{score}}}$, i.e., for which

$$\Pr[\|s(x) - s^*(x)\| > \varepsilon/\sigma] \ge \delta_{\text{score}}. \qquad (13)$$

**An intermediate notion.** In order to show equation 12, we take two steps. First, we separate out the choice of $x$, and that of $(z \mid x)$; we consider an intermediate measure that is the *empirical* average over $x$, and the *true* average over $(z \mid x)$. Since $\mathbb{E}_{z|x}[\Delta] = 0$, this is just the empirical $L^2$ error. Second, in order to limit the contribution of outliers we cannot reliably sample, we cap the contribution to $\frac{10B^2}{\sigma^2}$. That is, we define

$$A_x := \hat{\mathbb{E}}_x[\min(\mathbb{E}_{z|x}[l'(s, x, z)], \frac{10B^2}{\sigma^2})] = \hat{\mathbb{E}}_x[\min(\|s(x) - s^*(x)\|^2, \frac{10B^2}{\sigma^2})]. \qquad (14)$$

Under equation 13, for $m > O(\frac{\log \frac{1}{\delta_{\text{train}}}}{\delta_{\text{score}}})$, we will with $1 - \delta_{\text{train}}$ probability have

$$A \gtrsim \frac{\varepsilon^2 \delta_{\text{score}}}{\sigma^2}. \qquad (15)$$

**Concentration about the intermediate notion.** Finally, we show that for every set of samples $x_i$ satisfying equation 15, the empirical average over $z$ is with high probability at least half the true average over $z$, i.e., $A/2$. For each sample $x$, we split our analysis of $\hat{E}_{z|x}[l(s, x, z)]$ into two cases:

If $\|s(x) - s^*(x)\| > O(\frac{B}{\sigma})$, then by Cauchy-Schwarz and the assumption that $\|\Delta\| \leq 2B/\sigma$,

$$l'(s, x, z) \geq \|s(x) - s^*(x)\|^2 - O(\frac{B}{\sigma}) \|s(x) - s^*(x)\| \geq \frac{10B^2}{\sigma^2}$$

so these $x$ will contribute the maximum possible value to $A_x$, regardless of $z$ (in its bounded range). On the other hand, if $\|s(x) - s^*(x)\| < O(\frac{B}{\sigma})$, then $|l'(s, x, z)| \lesssim B^2/\sigma^2$ and

$$\text{Var}_{z|x}(l'(s, x, z)) = 4 \mathbb{E}[\langle s(x) - s^*(x), \Delta \rangle^2] \lesssim \frac{B^2}{\sigma^2} \|s(x) - s^*(x)\|^2$$

so for these $x$, as a distribution over $z$, $l'$ is bounded with bounded variance.

In either case, the contribution to $A_x$ is bounded with bounded variance; this lets us apply Bernstein's inequality to show, if $m > O(\frac{B^2 \log \frac{1}{\delta_{\text{train}}}}{\sigma^2 A})$, for every $x$ we will have

$$\hat{\mathbb{E}}_z[l'(s, x, z)] \geq \frac{A}{2} > 0$$

with $1 - \delta_{\text{train}}$ probability.

**Conclusion.** Suppose $m > O(\frac{B^2 \log \frac{1}{\delta_{\text{train}}}}{\varepsilon^2 \delta_{\text{score}}})$. Then with $1 - \delta_{\text{train}}$ probability we will have equation 15; and conditioned on this, with $1 - \delta_{\text{train}}$ probability we will have $\hat{\mathbb{E}}_{x,z} > 0$. Hence this $m$ suffices to distinguish any candidate score $s$ that is far from $s^*$.

Then for finite hypothesis classes we can take the union bound, incurring a $\log |\mathcal{H}|$ loss. This gives Theorem 1.1.

## 3.2 SAMPLING

Recall that the reverse SDE in equation 2 is discretized using score estimates $\hat{s}_{T-t_k}$ for some $0 < t_k < T$ as follows:

$$\mathrm{d}x_{T-t} = (x_{T-t} + 2\hat{s}_{T-t_k}(x_{T-t_k})) \, \mathrm{d}t + \sqrt{2} \, \mathrm{d}B_t.$$

Let $\hat{Q}$ be the law of the above approximate process, and let $Q$ be the law of the true reverse SDE in equation 2. Our goal is to bound $\mathsf{TV}(\hat{Q}, Q)$, the error incurred from following the above approximate SDE rather than the true one.

There are three sources of error: (a) Score estimation error, from making use of our estimated scores $\hat{s}_{t_k}$ rather than the true scores $s_{t_k}$. (b) Discretization error. (c) Initialization error, since $x_T \sim \mathcal{N}(0, I_d)$ rather than $q_T$. To bound (b) and (c), we make use of prior results (Benton et al., 2023). So, our main technical result is to bound the contribution of the score estimation error in our new $D_q^\delta$ sense to the final sampling error. Formally, we show the following.

**Lemma 3.1** (Main Sampling Lemma). *Consider an arbitrary sequence of discretization times* $0 = t_0 < t_1 < \cdots < t_N = T - \gamma$. *Assume that for each* $k \in \{0, \dots, N-1\}$, *the following holds:*

$$D_{q_{T-t_k}}^{\delta/N}(\widehat{s}_{T-t_k}, s_{T-t_k}) \leq \frac{\varepsilon}{\sigma_{T-t_k}} \cdot \frac{1}{\sqrt{T + \log \frac{1}{\gamma}}}$$

*Then, the output distribution* $\widehat{q}_{T-t_N}$ *satisfies*

$$\mathsf{TV}(\widehat{q}_{T-t_N}, q_{T-t_N}) \lesssim \delta + \varepsilon + \mathsf{TV}(Q, Q_{dis}) + \mathsf{TV}(q_T, \mathcal{N}(0, I_d))$$

In the above lemma, $Q_{\text{dis}}$ is a discretized version of the true reverse SDE in equation 2, using the true scores. Note that the assumption in the above lemma is satisfied for score estimates using the score-matching ERM using a number of samples scaling polynomially in $N$, via theorem 1.1. Per (Benton et al., 2023), $N$ scales polylogarithmically in $\frac{1}{\gamma}$. Also, as stated above, we can bound $\mathsf{TV}(Q, Q_{\text{dis}})$ and $\mathsf{TV}(q_T, \mathcal{N}(0, I_d))$ using previous results from (Benton et al., 2023). So, we will focus sketching the proof of the above – for the full proof of Theorem 1.2, see Appendix B.

To show the above, first, note that the error incurred from beginning at $\mathcal{N}(0, I_d)$ instead of $q_T$ is exactly $\mathsf{TV}(q_T, \mathcal{N}(0, I_d))$. For the remaining sketch, we will ignore this term, by assuming that all the processes start at $q_T$.

Observe that by the definition of $D_p^\delta$ from equation 8, a simple union bound yields that with probability $1 - \delta$ under $Q$, the score estimates satisfy

$$\sum_{k=0}^{N-1} \|\widehat{s}_{T-t_k}(x_{T-t_k}) - s_{T-t_k}(x_{T-t_k})\|^2 (t_{k+1} - t_k) \leq \varepsilon^2. \tag{16}$$

Now, for each $k \in \{0, \dots, N-1\}$, we will let $E_k$ be the event that the accumulated score estimation error up to time $T - t_k$ is at most $\varepsilon^2$. That is

$$E_k = \mathbf{1}\left\{ \sum_{i=0}^{k} \|\widehat{s}_{T-t_i}(x_{T-t_i}) - s_{T-t_i}(x_{T-t_i})\|^2 (t_{i+1} - t_i) \leq \varepsilon^2 \right\}.$$

Define the process $\widetilde{Q}$ initialized at $x_T \sim q_T$ that makes use of the *true* (discretized) score $s_{T-t_k}$ while the accumulated score error is at most $\varepsilon^2$, and then switches over to the score estimate $\widehat{s}_{T-t_k}$. That is, $x_T \sim q_T$, and for $t \in [t_k, t_{k+1}]$,

$$\mathrm{d}x_{T-t} = -(x_{T-t} + 2\widetilde{s}_{T-t_k}(x_{T-t_k}))\,\mathrm{d}t + \sqrt{2}\,\mathrm{d}B_t$$

where

$$\widetilde{s}_{T-t_k}(x_{T-t_k}) := \begin{cases} s_{T-t_k}(x_{T-t_k}) & E_k \text{ holds,} \\ \widehat{s}_{T-t_k}(x_{T-t_k}) & E_k \text{ doesn't hold.} \end{cases}$$

Now, by the triangle inequality, $\mathsf{TV}(Q, \widehat{Q}) \lesssim \mathsf{TV}(Q, \widetilde{Q}) + \mathsf{TV}(\widetilde{Q}, \widehat{Q})$. We will bound each term separately.

**Bounding $\mathsf{TV}(Q, \widetilde{Q})$.** With probability $1 - \delta$ over $Q$, all $E_k$ hold, since the total score estimation error is at most $\varepsilon^2$, per equation 16. So, with probability $1 - \delta - \mathsf{TV}(Q, Q_{\text{dis}})$ over $Q_{\text{dis}}$, all $E_k$ hold, where $Q_{\text{dis}}$ is the discretized process using the *true* scores. But, under this condition, $Q_{\text{dis}}$ and $\widetilde{Q}$ are the same process. So, $\mathsf{TV}(Q, \widetilde{Q}) \leq \delta + \mathsf{TV}(Q, Q_{\text{dis}})$.

**Bounding $\mathsf{TV}(\widetilde{Q}, \widehat{Q})$.** Here, we can apply Girsanov's theorem, which shows

$$D_{\mathrm{KL}}(\widetilde{Q} \| \widehat{Q}) \lesssim \mathbb{E}_{\widetilde{Q}}\left[ \sum_{k=0}^{N-1} \|\widetilde{s}_{T-t_k}(x_{T-t_k}) - \widehat{s}_{T-t_k}(x_{T-t_k})\|^2 (t_{k+1} - t_k) \right] \lesssim \varepsilon^2.$$

So, $\mathsf{TV}(\widetilde{Q}, \widehat{Q}) \lesssim \varepsilon$.

## 4   HARDNESS OF LEARNING IN $L^2$

In this section, we give concrete examples where it is information-theoretically hard to learn the score in $L^2$, and demonstrate that our proposed error measure can be much smaller than the $L^2$ error. In particular, as in equation (7), previous works require the $L^2$ error of the score estimate to be bounded. We show that this guarantee is prohibitively expensive to achieve – if $q_0$ has second moment 1, achieving the bound in (7) requires a number of samples polynomial in $\frac{1}{\sigma_t}$ to learn an estimate of $s_t$. So, to sample from a distribution $\gamma$-close in 2-Wasserstein to $q_0$, we would need polynomially many samples in $\frac{1}{\gamma}$.

For our first example, consider the two distributions $(1 - \eta)\mathcal{N}(0, 1) + \eta\mathcal{N}(\pm R, 1)$, where $R$ is polynomially large. Even though these distributions are polynomially bounded, it is *information-theoretically impossible* to distinguish these in $L^2$ given significantly fewer than $\frac{1}{\eta}$ samples. However, the $L^2$ error in score incurred from picking the score of the wrong distribution is large – polynomial in $R$. In Figure 1, the rightmost plot shows a simulation of this example, and demonstrates that the $L^2$ error remains large even after many samples are taken. Formally, we have:

**Lemma 4.1.** *Let $R$ be sufficiently large. Let $p_1$ be the distribution $(1 - \eta)\mathcal{N}(0, 1) + \eta\mathcal{N}(-R, 1)$ with corresponding score function $s_1$, and let $p_2$ be $(1 - \eta)\mathcal{N}(0, 1) + \eta\mathcal{N}(R, 1)$ with score $s_2$. Then, given $m$ samples from either distribution, it is impossible to distinguish between $p_1$ and $p_2$ for $\eta < \frac{1}{m^{1.1}}$ with probability larger than $1/2 + o_m(1)$. But,*

$$\mathbb{E}_{x \sim p_1}\left[\|s_1(x) - s_2(x)\|^2\right] \gtrsim \eta R^2 \quad and \quad \mathbb{E}_{x \sim p_2}\left[\|s_1(x) - s_2(x)\|^2\right] \gtrsim \eta R^2.$$

In the above example, the true distribution that our samples are drawn from is somewhat complex – a mixture of Gaussians. In the following example, we show that even if our true distribution is very simple–just a standard normal distribution, the score can still not be learned in $L^2$ definitively if the hypothesis class is large enough, for instance, in the case of neural networks.

In particular, let $\widehat{s}$ be the score of the mixture distribution $\eta\mathcal{N}(0, 1) + (1 - \eta)\mathcal{N}(S, 1)$, as in Figure 2. This score will have practically the same score matching objective as the true score for the given samples with high probability, as shown in Figure 2, since all $m$ samples will occur in the region where the two scores are nearly identical. However, the squared $L^2$ error incurred from picking the wrong score function $\widehat{s}$ is large – $\Omega\left(\frac{S^2}{m}\right)$. We formally state this result in the following lemma:

**Lemma 4.2.** *Let $S$ be sufficiently large. Consider the distribution $\widehat{p} = \eta\mathcal{N}(0, 1) + (1 - \eta)\mathcal{N}(S, 1)$ for $\eta = \frac{Se^{-\frac{S^2}{2} + 10\sqrt{\log m} \cdot S}}{10\sqrt{\log m}}$, and let $\widehat{s}$ be its score function. Given $m$ samples from the standard Gaussian $p^* = \mathcal{N}(0, 1)$ with score function $s^*$, with probability at least $1 - \frac{1}{poly(m)}$,*

$$\widehat{\mathbb{E}}\left[\|\widehat{s}(x) - s^*(x)\|^2\right] \le e^{-O(S\sqrt{\log m})} \quad but \quad \mathbb{E}_{x \sim p^*}\left[\|\widehat{s}(x) - s^*(x)\|^2\right] \gtrsim \frac{S^2}{m}.$$

Together, these examples show that the score *cannot* be learned in $L^2$ with fewer than $\mathrm{poly}(R/\gamma)$ samples, motivating our $1 - \delta$ quantile error measure.

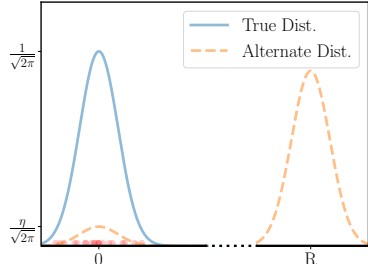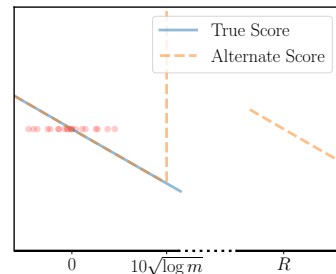

Figure 2: For $m$ samples from $\mathcal{N}(0, 1)$, consider the score $\widehat{s}$ of the mixture $\eta\mathcal{N}(0, 1) + (1 - \eta)\mathcal{N}(R, 1)$ above with $\eta$ is chosen so that $\widehat{s}(10\sqrt{\log m}) = 0$. For this $\widehat{s}$, the score-matching objective is close to 0, while the squared $L^2$ error is $\Omega\left(\frac{R^2}{m}\right)$.

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

## A  SCORE ESTIMATION

This section analyzes the score-matching objective for arbitrary distributions and bounded size function classes. Our main result (Theorem 1.1) shows that if there is a function in the function class $\mathcal{H}$ that approximates the score well in our $D_p^\delta$ sense (see equation 8), then the score-matching objective can learn this function using a number of samples that is independent of the domain size or the maximum value of the score.

We also show how to obtain a similar bound for the case when the function class $\mathcal{H}$ is the class of neural networks with bounded weight (Theorem 1.5).

A.1   SCORE ESTIMATION FOR FIXED TIME

**Notation.**   Fix a time $t$. For the purposes of this section, let $q := q_t$ be the distribution at time $t$, let $\sigma := \sigma_t$ be the smoothing level for time $t$, and let $s := s_t$ be the score function for time $t$. For $m$ samples $y_i \sim q_0$ and $z_i \sim \mathcal{N}(0, \sigma^2)$, let $x_i = e^{-t}y_i - z_i \sim q_t$.

We now state the score matching algorithm.

---

**Algorithm 1** Empirical score estimation for $s$

---

**Input:**   Distribution $q_0$, $y_1, \ldots, y_m \sim q_0$, set of hypothesis score function $\mathcal{H} = \{\tilde{s}_i\}$, smoothing level $\sigma$.

  1. Take $m$ independent samples $z_i \sim N(0, \sigma^2 I_d)$, and let $x_i = e^{-t}y_i - z_i$.
  2. For each $\tilde{s} \in \mathcal{H}$, let

$$l(\tilde{s}) = \frac{1}{m} \sum_{i=1}^{m} \left\| \tilde{s}(x_i) - \frac{-z_i}{\sigma^2} \right\|_2^2$$

  3. Let $\hat{s} = \arg\min_{\tilde{s} \in \mathcal{H}} l(\tilde{s})$
  4. Return $\hat{s}$

---

**Theorem 1.1** (Score Estimation for Finite Function Class). *For any distribution $q_0$ and time $t > 0$, consider the $\sigma_t$-smoothed version $q_t$ with associated score $s_t$. For any finite set $\mathcal{H}$ of candidate score functions. If there exists some $s^* \in \mathcal{H}$ such that*

$$\mathbb{E}_{x \sim q_t} \left[ \|s^*(x) - s_t(x)\|_2^2 \right] \leq \frac{\delta_{score} \cdot \delta_{train} \cdot \varepsilon^2}{100 \cdot \sigma_t^2}, \tag{9}$$

*for a sufficiently large constant $C$, then using $m > \widetilde{O}\left( \frac{1}{\varepsilon^2 \delta_{score}} (d + \log \frac{1}{\delta_{train}}) \log \frac{|\mathcal{H}|}{\delta_{train}} \right)$ samples, the empirical minimizer $\hat{s}$ of the score matching objective as described in equation 4 used to estimate $s_t$ satisfies*

$$D_{q_t}^{\delta_{score}}(\hat{s}, s_t) \leq \varepsilon / \sigma_t$$

*with probability $1 - \delta_{train}$.*

*Proof.*  Per the notation discussion above, we set $s = s_t$ and $\sigma = \sigma_t$.

Denote

$$l(s, x, z) := \left\| s(x) - \frac{z}{\sigma^2} \right\|_2^2$$

We will show that for all $\tilde{s}$ such that $D_q^{\delta_{score}}(\tilde{s}, s) > \varepsilon / \sigma$, with probability $1 - \delta_{train}$,

$$\hat{\mathbb{E}} \left[ l(\tilde{s}, x, z) - l(s^*, x, z) \right] > 0,$$

so that the empirical minimizer $\hat{s}$ is guaranteed to have

$$D_q^{\delta_{score}}(\hat{s}, s) \leq \varepsilon / \sigma.$$

We have

$$l(\tilde{s}, x, z) - l(s^*, x, z) = \left\| \tilde{s}(x) - \frac{z}{\sigma^2} \right\|^2 - \left\| s^*(x) - \frac{z}{\sigma^2} \right\|^2$$

$$= \left( \left\| \tilde{s}(x) - \frac{z}{\sigma^2} \right\|^2 - \left\| s(x) - \frac{z}{\sigma^2} \right\|^2 \right) - \|s^*(x) - s(x)\|^2 - 2\langle s^*(x) - s(x), s(x) - \frac{z}{\sigma^2} \rangle. \tag{17}$$

Note that by Markov's inequality, with probability $1 - \delta_{train}/3$,

$$\widehat{\mathbb{E}}_x \left[ \|s^*(x) - s(x)\|^2 \right] \leq \frac{\delta_{score} \cdot \varepsilon^2}{30 \cdot \sigma^2}.$$

Moreover, $s(x) = \mathbb{E}_{z|x}\left[\frac{z}{\sigma^2}\right]$ so that

$$\mathbb{E}\left[\langle s^*(x) - s(x), s(x) - \frac{z}{\sigma^2}\rangle\right] = 0$$

and

$$\mathbb{E}\left[\langle s^*(x) - s(x), s(x) - \frac{z}{\sigma^2}\rangle^2\right] \le \mathbb{E}_x\left[\|s^*(x) - s(x)\|^2\right] \mathbb{E}_{x,z}\left[\|s(x) - \frac{z}{\sigma^2}\|^2\right]$$

$$\le \frac{\delta_{\text{score}} \cdot \delta_{\text{train}} \cdot \varepsilon^2}{25\sigma^2} \cdot \frac{d}{\sigma^2}$$

So, by Chebyshev's inequality, with probability $1 - \delta_{\text{train}}/3$,

$$\widehat{\mathbb{E}}\left[\langle s^*(x) - s(x), s(x), \frac{z}{\sigma^2}\rangle\right] \lesssim \frac{1}{\sigma^2}\sqrt{\frac{\delta_{\text{score}} \cdot \varepsilon^2 \cdot d}{25m}} \le \frac{\delta_{\text{score}} \cdot \varepsilon^2}{100\sigma^2}$$

for our choice of $m$.

Also, by Corollary A.2, with probability $1 - \delta_{\text{train}}/3$, for all $\tilde{s} \in \mathcal{H}$ that satisfy $D_q^{\delta_{\text{score}}}(\tilde{s}, s) > \varepsilon/\sigma$ simultaneously,

$$\widehat{\mathbb{E}}\left[\left\|\tilde{s}(x) - \frac{z}{\sigma^2}\right\|^2 - \left\|s(x) - \frac{z}{\sigma^2}\right\|^2\right] \ge \frac{\delta_{\text{score}}\varepsilon^2}{16\sigma^2}.$$

Plugging in everything into equation equation 17, we have, with probability $1 - \delta_{\text{train}}$, for all $\tilde{s} \in \mathcal{H}$ with $D_q^{\delta_{\text{score}}}(\tilde{s}, s) > \varepsilon/\sigma$ simultaneously,

$$\widehat{\mathbb{E}}\left[l(\tilde{s}, x, z) - l(s^*, x, z)\right] \ge \frac{\delta_{\text{score}}\varepsilon^2}{16\sigma^2} - \frac{\delta_{\text{score}}\varepsilon^2}{100\sigma^2} - \frac{\delta_{\text{score}}\varepsilon^2}{100\sigma^2} > 0$$

as required. $\qquad\square$

**Lemma A.1.** *Consider any set $\mathcal{F}$ of functions $f : \mathbb{R}^d \to \mathbb{R}^d$ such that for all $f \in \mathcal{F}$,*

$$\Pr_{x \sim p}\left[\|f(x)\| > \varepsilon/\sigma\right] > \delta_{\text{score}}.$$

*Then, with $m > \widetilde{O}\left(\frac{1}{\varepsilon^2 \delta_{\text{score}}}(d + \log\frac{1}{\delta_{\text{train}}})\log\frac{|\mathcal{F}|}{\delta_{\text{train}}}\right)$ samples drawn in Algorithm 1, we have with probability $1 - \delta_{\text{train}}$,*

$$\frac{1}{m}\sum_{i=1}^{m} -2\left(\frac{z_i}{\sigma^2} - \mathbb{E}\left[\frac{z}{\sigma^2}|x_i\right]\right)^T f(x_i) + \frac{1}{2}\|f(x_i)\|^2 \ge \frac{\delta_{\text{score}} \cdot \varepsilon^2}{16\sigma^2}$$

*holds for all $f \in \mathcal{F}$.*

*Proof.* Define

$$h_f(x, z) := -2\left(\frac{z}{\sigma^2} - \mathbb{E}\left[\frac{z}{\sigma^2}|x\right]\right)^T f(x) + \frac{1}{2}\|f(x)\|^2$$

We want to show that $h_f$ has

$$\widehat{\mathbb{E}}\left[h_f(x, z)\right] := \frac{1}{m}\sum_{i=1}^{m} h_f(x_i, z_i) \ge \frac{\delta_{\text{score}}\varepsilon^2}{16\sigma^2} \tag{18}$$

for all $f \in \mathcal{F}$ with probability $1 - \delta_{\text{train}}$.

Let $B = O\left(\frac{\sqrt{d + \log\frac{m}{\varepsilon\delta_{\text{score}}\delta_{\text{train}}}}}{\sigma}\right)$. For $f \in \mathcal{F}$, let

$$g_f(x, z) = \begin{cases} B^2 & \text{if } \|f(x)\| \ge 10B \\ h_f(x, z) & \text{otherwise} \end{cases}$$

be a clipped version of $h_f(x, z)$. We will show that for our chosen number of samples $m$, the following hold with probability $1 - \delta_{\text{train}}$ simultaneously:

1. For all $i$, $\left\|\frac{z_i}{\sigma^2}\right\| \leq B$.

2. For all $i$, $\left\|\mathbb{E}[\frac{z}{\sigma^2}|x_i]\right\| \leq B$

3. $\hat{\mathbb{E}}[g_f(x,z)] \geq \frac{\delta_{\text{score}}\varepsilon^2}{16}$ for all $f \in \mathcal{F}$

To show that these together imply equation 18, note that whenever $g_f(x_i, z_i) \neq h_f(x_i, z_i)$, $\|f(x_i)\| \geq 10B$. So, since $\left\|\frac{z_i}{\sigma^2}\right\| \leq B$ and $\left\|\mathbb{E}[\frac{z}{\sigma^2}|x_i]\right\| \leq B$,

$$h_f(x_i, z_i) = -2\left(\frac{z_i}{\sigma^2} - \mathbb{E}\left[\frac{z}{\sigma^2}|x_i\right]\right)^T f(x_i) + \frac{1}{2}\|f(x_i)\|^2 \geq -4B\|f(x_i)\| + \frac{1}{2}\|f(x_i)\|^2 \geq B^2 \geq g_f(x_i, z_i).$$

So under conditions $1, 2, 3$, for all $f \in \mathcal{F}$,

$$\hat{\mathbb{E}}[h_f(x,z)] \geq \hat{\mathbb{E}}[g_f(x,z)] \geq \frac{\delta_{\text{score}}\varepsilon^2}{16\sigma^2}$$

So it just remains to show that conditions $1, 2, 3$ hold with probability $1 - \delta_{\text{train}}$ simultaneously.

1. **For all i, $\left\|\frac{z_i}{\sigma^2}\right\| \leq B$.** Holds with probability $1 - \delta_{\text{train}}/3$ by Lemma E.5 and the union bound.

2. **For all i, $\left\|\mathbb{E}\left[\frac{z}{\sigma^2} \mid x_i\right]\right\| \leq B$.** Holds with probability $1 - \delta_{\text{train}}/3$ by Lemma E.6 and the union bound.

3. $\hat{\mathbb{E}}[g_f(x,z)] \geq \frac{\delta_{\text{score}}\varepsilon^2}{16\sigma^2}$ **for all $f \in \mathcal{F}$.**

   Let $E$ be the event that 1. and 2. hold. Let $a_i = \min(\|f(x_i)\|, 10B)$. We proceed in multiple steps.

   - Conditioned on $E$, $|g_f(x_i, z_i)| \lesssim B^2$.
     If $\|f(x_i)\| \geq 10B$, $|g_f(x_i, z_i)| = B^2$ by definition. On the other hand, when $\|f(x_i)\| < 10B$, since we condition on $E$,
     $$|g_f(x_i, z_i)| = |h_f(x_i, z_i)| = \left|-2\left(\frac{z_i}{\sigma^2} - \mathbb{E}\left[\frac{z}{\sigma^2}|x_i\right]\right)^T f(x_i) + \frac{1}{2}\|f(x_i)\|^2\right| \lesssim B^2$$

   - $\mathbb{E}\left[g_f(x_i, z_i)|E, a_i\right] \gtrsim a_i^2 - O(\delta_{\text{train}}B^2)$.
     First, note that by definition of $g_f(x,z)$, for $a_i = 10B$,
     $$\mathbb{E}\left[g_f(x_i, z_i)|a_i = 10B\right] = B^2$$

   Now, for $a_i < 10B$,
   $$\mathbb{E}\left[g_f(x_i, z_i)|a_i\right] = \mathbb{E}\left[h_f(x_i, z_i)|a_i\right]$$
   $$= \mathbb{E}_{x_i|\|f(x_i)\|=a_i}\left[\mathbb{E}_{z|x_i}\left[h_f(x_i, z)\right]\right]$$

   Now, note that
   $$\mathbb{E}_{z|x}\left[h_f(x,z)\right] = \frac{1}{2}\|f(x)\|^2$$

   So, for $a < 10B$
   $$\mathbb{E}\left[g_f(x_i, z_i)|a_i\right] = \frac{1}{2}a_i^2$$

   Now let $g_f^{\text{clip}}(x_i, z_i)$ be a clipped version of $g_f(x_i, z_i)$, clipped to $\pm CB^2$ for sufficiently large constant $C$. We have, by above,
   $$\mathbb{E}[g_f(x_i, z_i)|a_i, E] = \mathbb{E}[g_f^{\text{clip}}(x_i, z_i)|a_i, E]$$

   But,
   $$\mathbb{E}[g_f^{\text{clip}}(x_i, z_i)|a_i, E] \geq \mathbb{E}[g_f^{\text{clip}}(x_i, z_i)|a_i] - O(\delta_{\text{train}}B^2)$$
   $$\gtrsim a_i^2 - O(\delta_{\text{train}}B^2)$$

- $\operatorname{Var}(g_f(x_i, z_i)|a_i, E) \lesssim a_i^2 B^2$.

  For $a_i = 10B$, we have, by definition of $g_f(x, z)$,

  $$\operatorname{Var}(g_f(x_i, z_i)|a_i, E) \lesssim B^4 \lesssim a_i^2 B^2$$

  On the other hand, for $a_i < 10B$,

  $$\operatorname{Var}(g_f(x_i, z_i)|a_i, E) \leq \mathbb{E}\left[g_f(x_i, z_i)^2|a_i, E\right]$$
  $$= \mathbb{E}\left[\left(-2\left(\frac{z_i}{\sigma^2} - \mathbb{E}\left[\frac{z}{\sigma^2}|x_i\right]\right)^T f(x_i) + \frac{1}{2}\|f(x_i)\|^2\right)^2\right]$$
  $$\lesssim a_i^2 B^2$$

  by Cauchy-Schwarz.

- With probability $1 - \delta_{\text{train}}/3$, for all $f \in \mathcal{F}$, $\widehat{\mathbb{E}}[g_f(x_i, z_i)] \gtrsim \Omega\left(\frac{\varepsilon^2 \delta_{\text{score}}}{\sigma^2}\right)$

  Using the above, by Bernstein's inequality, with probability $1 - \delta_{\text{train}}/6$,

  $$\widehat{\mathbb{E}}\left[g_f(x_i, z_i)|a_i, E\right] \gtrsim \frac{1}{n}\sum_{i=1}^n a_i^2 - O(\delta_{\text{train}} B^2) - \frac{1}{n}B\sqrt{\sum_{i=1}^n a_i^2 \log\frac{1}{\delta_{\text{train}}}} - \frac{1}{n}B^2 \log\frac{1}{\delta_{\text{train}}}$$

  Now, note that since $\Pr_{x \sim p_\sigma}\left[\|f(x)\| > \varepsilon/\sigma\right] \geq \delta_{\text{score}}$, we have with probability $1 - \delta_{\text{train}}/6$, for $n > O\left(\frac{\log\frac{1}{\delta_{\text{train}}}}{\delta_{\text{score}}}\right)$

  $$\frac{1}{n}\sum_{i=1}^n a_i^2 \geq \Omega\left(\frac{\varepsilon^2 \delta_{\text{score}}}{\sigma^2}\right)$$

  So, for $n > O\left(\frac{B^2 \cdot \sigma^2 \log\frac{1}{\delta_{\text{train}}}}{\varepsilon^2 \delta_{\text{score}}}\right)$, we have, with probability $1 - \delta_{\text{train}}/3$,

  $$\widehat{\mathbb{E}}\left[g_f(x_i, z_i)|E\right] \gtrsim \Omega\left(\frac{\varepsilon^2 \delta_{\text{score}}}{\sigma^2}\right) - O(\delta_{\text{train}} B^2)$$

  Rescaling so that $\delta_{\text{train}} \leq O\left(\frac{\varepsilon^2 \delta_{\text{score}}}{\sigma^2 \cdot B^2}\right)$, for $n > O\left(\frac{B^2 \cdot \sigma^2 \log\frac{B^2 \cdot \sigma^2}{\varepsilon^2 \delta_{\text{score}} \delta_{\text{train}}}}{\varepsilon^2 \cdot \delta_{\text{score}}}\right)$, we have, with probability $1 - \delta_{\text{train}}/3$,

  $$\widehat{\mathbb{E}}\left[g_f(x_i, z_i)|E\right] \gtrsim \Omega\left(\frac{\varepsilon^2 \delta_{\text{score}}}{\sigma^2}\right)$$

  Combining with 1. and 2. gives the claim for a single $f \in \mathcal{F}$. Union bounding over the size of $\mathcal{F}$ gives the claim.

  $\square$

**Corollary A.2.** *Let $\mathcal{H}_{bad}$ be a set of score functions such that for all $\tilde{s} \in \mathcal{H}_{bad}$,*

$$D_q^{\delta_{score}}(\tilde{s}, s) > \varepsilon/\sigma.$$

*Then, for $m > \widetilde{O}\left(\frac{1}{\varepsilon^2 \delta_{score}}(d + \log\frac{1}{\delta_{train}})\log\frac{|\mathcal{H}_{bad}|}{\delta_{train}}\right)$ samples drawn by Algorithm 1, we have with probability $1 - \delta_{train}$,*

$$\widehat{\mathbb{E}}\left[\|\tilde{s}(x) - \frac{z}{\sigma^2}\|^2 - \|s(x) - \frac{z}{\sigma^2}\|^2\right] \geq \frac{\delta_{score}\varepsilon^2}{16\sigma^2}$$

*for all $\tilde{s} \in \mathcal{H}_{bad}$.*

*Proof.* We have, for $f(x) := \tilde{s}(x) - s(x)$,

$$\|\tilde{s}(x) - \frac{z}{\sigma^2}\|^2 - \|s(x) - \frac{z}{\sigma^2}\|^2 = \|f(x) + (s(x) - \frac{z}{\sigma^2})\|^2 - \|s(x) - \frac{z}{\sigma^2}\|^2$$

$$= \|f(x)\|^2 + 2(s(x) - \frac{z}{\sigma^2})^T f(x)$$

$$= \|f(x)\|^2 - 2(\frac{z}{\sigma^2} - \mathbb{E}[\frac{z}{\sigma^2}|x])^T f(x)$$

since $s(x) = \mathbb{E}\left[\frac{z}{\sigma^2}|x\right]$ by Lemma E.1. Then, by definition, for $s \in \mathcal{H}_{\text{bad}}$, for the associated $f$, $\Pr[\|f(x)\| > \varepsilon/\sigma] > \delta_{\text{score}}$. So, by Lemma A.1, the claim follows. $\square$

## A.2 Score Estimation for Diffusion Process

**Theorem A.3.** *Let $q_0$ be a distribution over $\mathbb{R}^d$. For discretization times $0 = t_0 < t_1 < \cdots < t_N$, let $s_{T-t_k}$ be the true score function of $q_{T-t_k}$. If for each $t_k$, the function class $\mathcal{H}$ contains some $\tilde{s}_{T-t_k} \in \mathcal{H}$ with*

$$\mathop{\mathbb{E}}_{x \sim q_{T-t_k}} \left[\|\tilde{s}_{T-t_k}(x) - s_{T-t_k}(x)\|_2^2\right] \leq \frac{\delta_{score} \cdot \delta_{train} \cdot \varepsilon^2}{40\sigma_{T-t_k}^2 \cdot N^2} \tag{19}$$

*Then, if we take $m > \widetilde{O}\left(\frac{N}{\varepsilon^2 \delta_{score}}(d + \log \frac{1}{\delta_{train}}) \log \frac{|\mathcal{H}|}{\delta_{train}}\right)$ samples, then with probability $1 - \delta_{train}$, each score $\hat{s}_{T-t_k}$ learned by score matching satisfies*

$$D_{q_{T-t_k}}^{\delta_{score}/N}(\hat{s}_{T-t_k}, s_{T-t_k}) \leq \varepsilon/\sigma_{T-t_k}.$$

*Proof.* Note that for each $t_k$, $q_{T-t_k}$ is a $\sigma_{T-t_k}$-smoothed distribution. Therefore, we can use theorem 1.1 by taking $\delta_{\text{train}}/N$ into $\delta_{\text{train}}$ and taking $\delta_{\text{score}}/N$ into $\delta_{\text{score}}$. We have that for each $t_k$, with probability $1 - \delta_{\text{train}}/N$ the following holds:

$$\mathop{\Pr}_{x \sim q_{T-t_k}} \left[\|\hat{s}_{T-t_k}(x) - s^*_{T-t_k}(x)\| \leq \varepsilon/\sigma_{T-t_k}\right] \geq 1 - \delta_{\text{score}}/N.$$

By a union bound over all the steps, we conclude the proposed statement. $\square$

## A.3 Score Training for Neural Networks

**Theorem 1.5** (Score Training for Neural Networks). *For any distribution $q_0$ with second moment $m_2^2$, and any time $t > 0$, let $q_t$ be the $\sigma_t$-smoothed version with associated score $s_t$. Let $\phi_\theta(\cdot)$ be the fully connected neural network with ReLU activations parameterized by $\theta$, with $P$ total parameters and depth $D$. If there exists some weight vector $\theta^*$ with $\|\theta^*\|_F \leq \Theta$ such that*

$$\mathop{\mathbb{E}}_{x \sim q_t} \left[\|\phi_{\theta^*}(x) - s_t(x)\|^2\right] \leq \frac{\delta_{score} \cdot \delta_{train} \cdot \varepsilon^2}{1000 \cdot \sigma_t^2}$$

*then using $m > \widetilde{O}\left(\frac{(d + \log \frac{1}{\delta_{train}}) \cdot PD}{\varepsilon^2 \delta_{score}} \cdot \log\left(\frac{(m_2 + \sigma)\Theta}{\delta_{train}}\right)\right)$ samples, the empirical minimizer $\phi_{\hat{\theta}}$ of the score matching objective used to estimate $s_t$ (over $\phi_\theta$ with $\|\theta\|_F \leq \Theta$) satisfies*

$$D_{q_t}^{\delta_{score}}(\phi_{\hat{\theta}}, s_t) \leq \varepsilon/\sigma.$$

*with probability $1 - \delta_{train}$.*

*Proof.* Per the notation discussion above, we set $s = s_t$ and $\sigma = \sigma_t$.

For any function $f$ denote

$$l(f, x, z) := \left\|f(x) - \frac{z}{\sigma^2}\right\|^2$$

We will show that for every $\tilde{\theta}$ with $\|\tilde{\theta}\|_F \leq \Theta$ such that $D_q^{\delta_{\text{score}}}(\phi_{\tilde{\theta}}, s) > \varepsilon/\sigma$, with probability $1 - \delta_{\text{train}}$,

$$\widehat{\mathbb{E}}\left[l(\phi_{\tilde{\theta}}, x, z) - l(\phi_{\theta^*}, x, z)\right] > 0$$

so that the empirical minimizer $\phi_{\widehat{\theta}}$ is guaranteed to have

$$D_q^{\delta_{\text{score}}}(\phi_{\widehat{\theta}}, s) \leq \varepsilon/\sigma$$

First, note that since the ReLU activation is contractive, the total Lipschitzness of $\phi_\theta$ is at most the product of the spectral norm of the weight matrices at each layer. For any $\theta$, consider $\widetilde{\theta}$ such that

$$\|\widetilde{\theta} - \theta\|_F \leq \frac{\tau}{\sigma D \Theta^{D-1}}$$

Let $M_1, \ldots, M_D$ be the weight matrices at each layer of the neural net $\phi_\theta$, and let $\widetilde{M}_1, \ldots, \widetilde{M}_D$ be the corresponding matrices of $\phi_{\widetilde{\theta}}$.

We now show that $\|\phi_{\widetilde{\theta}}(x) - \phi_\theta(x)\|$ is small, using a hybrid argument. Define $y_i$ to be the output of a neural network with weight matrices $M_1, \ldots, M_i, \widetilde{M}_{i+1}, \ldots, \widetilde{M}_D$ on input $x$, so $y_0 = \phi_{\widetilde{\theta}}(x)$ and $y_D = \phi_\theta(x)$. Then we have

$$\|y_i - y_{i+1}\| \leq \|x\| \cdot \left(\prod_{j \leq i} \|M_j\|\right) \cdot \left\|M_{i+1} - \widetilde{M}_{i+1}\right\| \cdot \left(\prod_{j > i+1} \left\|\widetilde{M}_j\right\|\right)$$

$$\leq \|x\| \Theta^{D-1} \left\|\widetilde{\theta} - \theta\right\|_F$$

and so

$$\left\|\phi_{\widetilde{\theta}}(x) - \phi_\theta(x)\right\|_2 = \|y_0 - y_D\| \leq \sum_{i=0}^{D-1} \|y_i - y_{i+1}\| \leq \|x\| D \Theta^{D-1} \left\|\widetilde{\theta} - \theta\right\|_F \leq \|x\| \cdot \tau/\sigma.$$

Note that the dimensionality of $\theta$ is $P$. So, we can construct a $\frac{\tau}{\sigma D \Theta^{D-1}}$-net $N$ over the set $\{\theta : \|\theta\|_F \leq \Theta\}$ of size $O\left(\frac{\sigma D \Theta^{D-1}}{\tau}\right)^P$, so that for any $\theta$ with $\|\theta\|_F \leq \Theta$, there exists $\widetilde{\theta} \in N$ with

$$\|\phi_{\widetilde{\theta}}(x) - \phi_\theta(x)\|_2 \leq (\tau/\sigma) \cdot \|x\|$$

Let $\mathcal{H} = \{\phi_{\widetilde{\theta}} : \widetilde{\theta} \in N\}$. Then, we have that for every $\theta$ with $\|\theta\|_F \leq \Theta$, there exists $h \in \mathcal{H}$ such that

$$\widehat{\mathbb{E}}\left[\|h(x) - \phi_\theta(x)\|^2\right] \leq (\tau/\sigma)^2 \cdot \frac{1}{m} \sum_{i=1}^m \|x_i\|^2 \tag{20}$$

Now, choose any $\widetilde{\theta}$ with $\|\widetilde{\theta}\|_F \leq \Theta$ and $D_q^{\delta_{\text{score}}}(\phi_{\widetilde{\theta}}, s) > \varepsilon/\sigma$, and let $\widetilde{h} \in \mathcal{H}$ satisfy the above for $\widetilde{\theta}$.

Our final choice of $m$ will satisfy $m > \widetilde{O}\left(\frac{1}{\varepsilon^2 \delta_{\text{score}}}\left(d + \log \frac{1}{\delta_{\text{train}}}\right) \log \frac{|\mathcal{H}|}{\delta_{\text{train}}}\right)$

We have

$$\begin{aligned}
&l(\phi_{\widetilde{\theta}}, x, z) - l(\phi_{\theta^*}, x, z) \\
&= \|\phi_{\widetilde{\theta}}(x) - \frac{z}{\sigma^2}\|^2 - \|\phi_{\theta^*}(x) - \frac{z}{\sigma^2}\|^2 \\
&= \|\phi_{\widetilde{\theta}}(x) - \widetilde{h}(x)\|^2 + 2\langle \phi_{\widetilde{\theta}}(x) - \widetilde{h}(x), \widetilde{h}(x) - \frac{z}{\sigma^2}\rangle + \|\widetilde{h}(x) - \frac{z}{\sigma^2}\|^2 \\
&\quad - \|s(x) - \frac{z}{\sigma^2}\|^2 - \|\phi_{\theta^*}(x) - s(x)\|^2 - 2\langle \phi_{\theta^*}(x) - s(x), s(x) - \frac{z}{\sigma^2}\rangle
\end{aligned} \tag{21}$$

Now, by Corollary A.2, for our choice of $m$, with probability $1 - \delta_{\text{train}}/4$ for every $h \in \mathcal{H}$ with $D_q^{2\delta_{\text{score}}}(h, s) > \varepsilon/(2\sigma)$ simultaneously,

$$\widehat{\mathbb{E}}\left[\|h(x) - \frac{z}{\sigma^2}\|^2 - \|s(x) - \frac{z}{\sigma^2}\|^2\right] \geq \frac{\delta_{\text{score}} \varepsilon^2}{128\sigma^2}$$

By Markov's inequality, with probability $1 - \delta_{\text{train}}/4$,

$$\widehat{\mathbb{E}}\left[\|\phi_{\theta^*}(x) - s(x)\|^2\right] \leq \frac{\delta_{\text{score}} \cdot \varepsilon^2}{250\sigma^2}$$

Now, since $s(x) = \mathbb{E}_{z|x}\left[\frac{z}{\sigma^2}\right]$,

$$\mathbb{E}\left[\langle \phi_{\theta^*}(x) - s(x), s(x) - \frac{z}{\sigma^2}\rangle\right] = 0$$

and

$$\mathbb{E}\left[\langle \phi_{\theta^*}(x) - s(x), s(x) - \frac{z}{\sigma^2}\rangle^2\right] \leq \mathbb{E}_x\left[\|\phi_{\theta^*}(x) - s(x)\|^2\right] \cdot \mathbb{E}_{x,z}\left[\|s(x) - \frac{z}{\sigma^2}\|^2\right]$$

$$\leq \frac{\delta_{\text{score}} \cdot \delta_{\text{train}} \cdot \varepsilon^2}{250 \cdot \sigma^2} \cdot \frac{d}{\sigma^2}$$

So, by Chebyshev's inequality, with probability $1 - \delta_{\text{train}}/4$

$$\widehat{\mathbb{E}}\left[\langle \phi_{\theta^*}(x) - s(x), s(x) - \frac{z}{\sigma^2}\rangle\right] \leq \frac{1}{\sigma^2}\sqrt{\frac{\delta_{\text{score}} \cdot \varepsilon^2 \cdot d}{250m}} \leq \frac{\delta_{\text{score}} \cdot \varepsilon^2}{1000 \cdot \sigma^2}$$

for our choice of $m$. So, by the above and equation 21 we have shown that with probability $1 - 3\delta_{\text{train}}/4$, as long as $\widetilde{h}$ has $D_q^{2\delta_{\text{score}}}(\widetilde{h}, s) > \varepsilon/(2\sigma)$,

$$\widehat{\mathbb{E}}\left[l(\phi_{\widetilde{\theta}}, x, z) - l(\phi_{\theta^*}, x, z)\right] \geq \frac{\delta_{\text{score}} \cdot \varepsilon^2}{500 \cdot \sigma^2} + 2\langle \phi_{\widetilde{\theta}}(x) - \widetilde{h}(x), \widetilde{h}(x) - \frac{z}{\sigma^2}\rangle \tag{22}$$

Now, we will show that $D_q^{2\delta_{\text{score}}}(\widetilde{h}, s) > \varepsilon/(2\sigma)$, as well as bound the last term above.

By the fact that $q_0$ has second moment $m_2^2$, we have that with probability $1 - \delta$ over $x$,

$$\|x\| \leq \frac{m_2}{\sqrt{\delta}} + \sigma\left(\sqrt{d} + \sqrt{\log\frac{1}{\delta}}\right)$$

Now since $D_q^{\delta_{\text{score}}}(\phi_{\widetilde{\theta}}, s) > \varepsilon/\sigma$, and $\|\widetilde{h}(x) - \phi_{\widetilde{\theta}}(x)\| \leq (\tau/\sigma) \cdot \|x\|$, we have, with probability at least $1 - 2\delta_{\text{score}}$,

$$\|\widetilde{h}(x) - s(x)\| \geq \|\phi_{\widetilde{\theta}}(x) - s(x)\| - \|\widetilde{h}(x) - \phi_{\widetilde{\theta}}(x)\|$$

$$\geq \varepsilon/\sigma - (\tau/\sigma) \cdot \left(\frac{m_2}{\sqrt{\delta_{\text{score}}}} + \sigma\left(\sqrt{d} + \sqrt{\log\frac{1}{\delta_{\text{score}}}}\right)\right)$$

$$\geq \varepsilon/(2\sigma)$$

for $\tau < C\varepsilon\delta_{\text{score}}\delta_{\text{train}}\frac{1}{\Theta^D\left(m \cdot m_2^2 + \sigma^2(d + \log\frac{m}{\delta_{\text{train}}})\right)}$ for some small enough constant $C$. So, we have shown that $D_q^{2\delta_{\text{score}}}(\widetilde{h}, s) > \varepsilon/(2\sigma)$.

Finally, we bound the last term in equation 22 above. We have by equation 20 and a union bound, with probability $1 - \delta_{\text{train}}/8$,

$$\widehat{\mathbb{E}}\left[\|\widetilde{h}(x) - \phi_{\widetilde{\theta}}(x)\|^2\right] \lesssim (\tau/\sigma)^2 \cdot \left(\frac{m \cdot m_2^2}{\delta_{\text{train}}} + \sigma^2\left(d + \log\frac{m}{\delta_{\text{train}}}\right)\right)$$

for $\tau < C\varepsilon\delta_{\text{score}}\delta_{\text{train}}\frac{1}{\Theta^D\left(m \cdot m_2^2 + \sigma^2(d + \log\frac{m}{\delta_{\text{train}}})\right)}$ for some small enough constant $C$. Similarly, with probability $1 - \delta_{\text{train}}/8$,

$$\widehat{\mathbb{E}}\left[\|\widetilde{h}(x) - \frac{z}{\sigma^2}\|^2\right] \lesssim \Theta^D \cdot \left(\frac{m \cdot m_2^2}{\delta_{\text{train}}} + \sigma^2\left(d + \log\frac{m}{\delta_{\text{train}}}\right)\right) + \frac{1}{\sigma^2}\left(d + \log\frac{m}{\delta_{\text{train}}}\right)$$

So, with probability $1 - \delta_{\text{train}}/4$, for $\tau < C\varepsilon\delta_{\text{score}}\delta_{\text{train}}\frac{1}{\Theta^D\left(m \cdot m_2^2 + \sigma^2(d + \log\frac{m}{\delta_{\text{train}}})\right)}$ for some small enough constant $C$,

$$\widehat{\mathbb{E}}\left[\langle \phi_{\widetilde{\theta}}(x) - \widetilde{h}(x), \widetilde{h}(x) - \frac{z}{\sigma^2}\rangle\right] \geq -\widehat{\mathbb{E}}\left[\|\widetilde{h}(x) - \phi_{\widetilde{\theta}}(x)\|^2\right] \cdot \mathbb{E}\left[\|\widetilde{h}(x) - \frac{z}{\sigma^2}\|^2\right]$$

$$\geq -\frac{\delta_{\text{score}}\varepsilon^2}{2000 \cdot \sigma^2}$$

So finally, combining with equation 22, we have with probability $1 - \delta_{\text{train}}$

$$\widehat{\mathbb{E}}\left[l(\phi_{\widetilde{\theta}}, x, z) - l(\phi_{\theta^*}, x, z)\right] \geq \frac{\delta_{\text{score}} \cdot \varepsilon^2}{1000 \cdot \sigma^2} > 0$$

as required.

$\square$

## B  SAMPLING WITH OUR SCORE ESTIMATION GUARANTEE

In this section, we show that diffusion models can converge to the true distribution without necessarily adhering to an $L^2$ bound on the score estimation error. A high probability accuracy of the score is sufficient.

In order to simulate the reverse process of equation 1 in an actual algorithm, the time was discretized into $N$ steps. The $k$-th step ends at time $t_k$, satisfying $0 \leq t_0 < t_1 < \cdots < t_N = T - \gamma$. The algorithm stops at $t_N$ and outputs the final state $x_{T-t_N}$.

To analyze the reverse process run under different levels of idealness, we consider these four specific path measures over the path space $\mathcal{C}([0, T - \gamma]; \mathbb{R}^d)$:

- Let $Q$ be the measure for the process that

$$\mathrm{d}x_{T-t} = (x_{T-t} + 2s_{T-t}(x_{T-t}))\,\mathrm{d}t + \sqrt{2}\,\mathrm{d}B_t, \quad x_T \sim q_T.$$

- Let $Q_{\text{dis}}$ be the measure for the process that for $t \in [t_k, t_{k+1}]$,

$$\mathrm{d}x_{T-t} = (x_{T-t} + 2s_{T-t_k}(x_{T-t_k}))\,\mathrm{d}t + \sqrt{2}\,\mathrm{d}B_t, \quad x_T \sim q_T.$$

- Let $\overline{Q}$ be the measure for the process that for $t \in [t_k, t_{k+1}]$,

$$\mathrm{d}x_{T-t} = (x_{T-t} + 2\widehat{s}_{T-t_k}(x_{T-t_k}))\,\mathrm{d}t + \sqrt{2}\,\mathrm{d}B_t, \quad x_T \sim q_T.$$

- Let $\widehat{Q}$ be the measure for the process that for $t \in [t_k, t_{k+1}]$,

$$\mathrm{d}x_{T-t} = (x_{T-t} + 2\widehat{s}_{T-t_k}(x_{T-t_k}))\,\mathrm{d}t + \sqrt{2}\,\mathrm{d}B_t, \quad x_T \sim \mathcal{N}(0, I_d).$$

To summarize, $Q$ represents the perfect reverse process of equation 1, $Q_{\text{dis}}$ is the discretized version of $Q$, $\overline{Q}$ runs $Q_{\text{dis}}$ with an estimated score, and $\widehat{Q}$ starts $\overline{Q}$ at $\mathcal{N}(0, I_d)$ — effectively the actual implementable reverse process.

Recent works have shown that under the assumption that the estimated score function is close to the real score function in $L^2$, then the output of $\widehat{Q}$ will approximate the true distribution closely. Our next theorem shows that this assumption is in fact not required, and it shows that our score assumption can be easily integrated in a black-box way to achieve similar results.

**Lemma B.1** (Score Estimation guarantee). *Consider an arbitrary sequence of discretization times* $0 = t_0 < t_1 < \cdots < t_N = T - \gamma$, *and let* $\sigma_t := \sqrt{1 - e^{-2t}}$. *Assume that for each* $k \in \{0, \ldots, N-1\}$, *the following holds:*

$$D_{q_{T-t_k}}^{\delta/N}\left(\widehat{s}_{T-t_k}, s_{T-t_k}\right) \leq \frac{\varepsilon}{\sigma_{T-t_k}}.$$

*Then, we have*

$$\Pr_Q\left[\sum_{k=0}^{N-1}\|\widehat{s}_{T-t_k}(x_{T-t_k}) - s_{T-t_k}(x_{T-t_k})\|_2^2(t_{k+1} - t_k) \leq \varepsilon^2\left(T + \log\frac{1}{\gamma}\right)\right] \geq 1 - \delta.$$

*Proof.* Since random variable $x_{T-t_k}$ follows distribution $q_{T-t_k}$ under $Q$, for each $k \in \{0, \ldots, N-1\}$, we have

$$\Pr_Q \left[ \|\widehat{s}_{T-t_k}(x_{T-t_k}) - s_{T-t_k}(x_{T-t_k})\| \leq \frac{\varepsilon}{\sqrt{1 - e^{-2(T-t_k)}}} \right] \geq 1 - \frac{\delta}{N}.$$

Using a union bound over all $N$ different $\sigma$ values, it follows that with probability at least $1 - \delta$ over $Q$, the inequality

$$\|\widehat{s}_{T-t_k}(x_{T-t_k}) - s_{T-t_k}(x_{T-t_k})\|_2^2 \leq \frac{\varepsilon^2}{1 - e^{-2(T-t_k)}}.$$

is satisfied for every $k \in \{0, \ldots, N-1\}$. Under this condition, we have

$$\sum_{k=0}^{N-1} \|\widehat{s}_{T-t_k}(x_{T-t_k}) - s_{T-t_k}(x_{T-t_k})\|_2^2 (t_{k+1} - t_k)$$

$$\leq \sum_{k=0}^{N-1} \frac{\varepsilon^2}{1 - e^{-2(T-t_k)}}(t_{k+1} - t_k)$$

$$\leq \sum_{k=0}^{N-1} \int_{t_k}^{t_{k+1}} \frac{\varepsilon^2}{1 - e^{-2(T-t_k)}} \, \mathrm{d}t$$

$$\leq \int_0^{T-\gamma} \frac{\varepsilon^2}{1 - e^{-2(T-t_k)}} \, \mathrm{d}t$$

$$\leq \varepsilon^2 \left( T + \log \frac{1}{\gamma} \right).$$

Hence, we find that

$$\Pr_Q \left[ \sum_{k=0}^{N-1} \|\widehat{s}_{T-t_k}(x_{T-t_k}) - s_{T-t_k}(x_{T-t_k})\|_2^2 (t_{k+1} - t_k) \leq \varepsilon^2 \left( T + \log \frac{1}{\gamma} \right) \right] \geq 1 - \delta.$$

$\square$

**Lemma B.2** (Score estimation error to TV). *Let $q$ be an arbitrary distribution. If the score estimation satisfies that*

$$\Pr_Q \left[ \sum_{k=0}^{N-1} \|\widehat{s}_{T-t_k}(x_{T-t_k}) - s_{T-t_k}(x_{T-t_k})\|_2^2 (t_{k+1} - t_k) \leq \varepsilon^2 \right] \geq 1 - \delta, \tag{23}$$

*then the output distribution $p_{T-t_N}$ of $\widehat{Q}$ satisfies*

$$\mathsf{TV}(q_\gamma, p_{T-t_N}) \lesssim \delta + \varepsilon + \mathsf{TV}(Q, Q_{dis}) + \mathsf{TV}(q_T, \mathcal{N}(0, I_d)).$$

*Proof.* We will start by bounding the TV distance between $Q_{\text{dis}}$ and $\overline{Q}$. We will proceed by defining $\widetilde{Q}$ and arguing that both $\mathsf{TV}(Q_{\text{dis}}, \widetilde{Q})$ and $\mathsf{TV}(\widetilde{Q}, \overline{Q})$ are small. By the triangle inequality, this will imply that $Q$ and $\overline{Q}$ are close in TV distance.

**Defining $\widetilde{Q}$.** For $k \in \{0, \ldots, N-1\}$, consider event

$$E_k := \left( \sum_{i=0}^{k} \|\widehat{s}_{T-t_i}(x_{T-t_i}) - s_{T-t_i}(x_{T-t_i})\|_2^2 (t_{i+1} - t_i) \leq \varepsilon^2 \right),$$

which represents that the accumulated score estimation error along the path is at most $\varepsilon^2$ for a discretized diffusion process.

Given $E_k$, we define a version of $Q_{\text{dis}}$ that is forced to have a bounded score estimation error. Let $\widetilde{Q}$ over $\mathcal{C}((0, T], \mathbb{R}^d)$ be the law of a modified reverse process initialized at $x_T \sim q_T$, and for each $t \in [t_k, t_{k+1})$,

$$\mathrm{d}x_{T-t} = -\left(x_{T-t} + 2\widetilde{s}_{T-t_k}(x_{T-t_k})\right) \mathrm{d}t + \sqrt{2} \, \mathrm{d}B_t, \tag{24}$$

where

$$\widetilde{s}_{T-t_k}(x_{T-t_k}) := \begin{cases} s_{T-t_k}(x_{T-t_k}) & E_k \text{ holds,} \\ \widehat{s}_{T-t_k}(x_{T-t_k}) & E_k \text{ doesn't hold.} \end{cases}$$

This SDE guarantees that once the accumulated score error exceeds $\varepsilon_{score}^2$ ($E_k$ fails to hold), we switch from the true score to the estimated score. Therefore, we have that the following inequality always holds:

$$\sum_{k=0}^{N-1} \|\widetilde{s}_{T-t_k}(x_{T-t_k}) - \widehat{s}_{T-t_k}(x_{T-t_k})\|_2^2 (t_{k+1} - t_k) \le \varepsilon^2. \tag{25}$$

$Q_{\text{dis}}$ **and** $\widetilde{Q}$ **are close.** By (23), we have

$$\Pr_{Q_{\text{dis}}} [E_0 \wedge \cdots \wedge E_{N-1}] = \Pr_{Q_{\text{dis}}} [E_{N-1}] \ge \Pr_{Q} [E_{N-1}] - \mathsf{TV}(Q, Q_{\text{dis}}) \ge 1 - \delta - \mathsf{TV}(Q, Q_{\text{dis}}),$$

Note that when a path $(x_{T-t})_{t \in [0, t_N]}$ satisfies $E_0 \wedge \cdots \wedge E_{N-1}$, its probability under $\widetilde{Q}$ is at least its probability under $Q_{\text{dis}}$. Therefore, we have

$$\mathsf{TV}(Q_{\text{dis}}, \widetilde{Q}) \lesssim \delta + \mathsf{TV}(Q, Q_{\text{dis}}).$$

$\widetilde{Q}$ **and** $\overline{Q}$ **are close.** Inspired by Chen et al., we utilize Girsanov's theorem (see theorem E.7) to help bound this distance. Define

$$b_r := \sqrt{2}(\widetilde{s}_{T-t_k}(x_{T-t_k}) - \widehat{s}_{T-t_k}(x_{T-t_k})),$$

where $k$ is index such that $r \in [t_k, t_{k+1})$. We apply the Girsanov's theorem to $(\widetilde{Q}, (b_r))$. By eq. (25), we have

$$\int_0^{t_N} \|b_r\|_2^2 \, \mathrm{d}r \le \sum_{k=0}^{N-1} \|\sqrt{2}(\widetilde{s}_{T-t_k}(x_{T-t_k}) - \widehat{s}_{T-t_k}(x_{T-t_k}))\|_2^2 (t_{k+1} - t_k) \le 2\varepsilon^2 < \infty.$$

This satisfies Novikov's condition and tells us that for

$$\mathcal{E}(\mathcal{L})_t = \exp\left( \int_0^t b_r \, \mathrm{d}B_r - \frac{1}{2} \int_0^t \|b_r\|_2^2 \, \mathrm{d}r \right),$$

under measure $\widetilde{Q}' := \mathcal{E}(\mathcal{L})_{t_N} \widetilde{Q}$, there exists a Brownian motion $(\widetilde{B}_t)_{t \in [0, t_N]}$ such that

$$\widetilde{B}_t = B_t - \int_0^t b_r \, \mathrm{d}r,$$

and thus for $t \in [t_k, t_{k+1})$,

$$\mathrm{d}\widetilde{B}_t = \mathrm{d}B_t + \sqrt{2}(\widetilde{s}_{T-t_k}(x_{T-t_k}) - \widehat{s}_{T-t_k}(x_{T-t_k})) \, \mathrm{d}t.$$

Plug this into (24) and we have that for $t \in [t_k, t_{k+1})$

$$\mathrm{d}x_{T-t} = -(x_{T-t} + 2\widehat{s}_{T-t_k}(x_{T-t_k})) \, \mathrm{d}t + \sqrt{2} \, \mathrm{d}\widetilde{B}_t, \quad x_T \sim q_T.$$

This equation depicts the distribution of $x$, and this exactly matches the definition of $\overline{Q}$. Therefore, $\overline{Q} = \widetilde{Q}' = \mathcal{E}(\mathcal{L})_{t_N} \widetilde{Q}$, and we have

$$D_{\mathrm{KL}}\left(\widetilde{Q} \big\| \overline{Q}\right) = \mathbb{E}_{\widetilde{Q}}\left[ \ln \frac{\mathrm{d}\widetilde{Q}}{\mathrm{d}\overline{Q}} \right] = \mathbb{E}_{\widetilde{Q}}\left[ \ln \mathcal{E}(\mathcal{L})_{t_N} \right].$$

Then by using (25), we have

$$\mathbb{E}_{\widetilde{Q}}\left[ \ln \mathcal{E}(\mathcal{L})_{t_N} \right] \lesssim \mathbb{E}_{\widetilde{Q}}\left[ \sum_{k=0}^{N-1} \|\widetilde{s}_{T-t_k}(x_{T-t_k}) - \widehat{s}_{T-t_k}(x_{T-t_k})\|_2^2 (t_{k+1} - t_k) \right] \lesssim \varepsilon^2.$$

Therefore, we can apply Pinsker's inequality and get

$$\mathsf{TV}(\widetilde{Q}, \overline{Q}) \le \sqrt{D_{\mathrm{KL}}\left(\widetilde{Q} \big\| \overline{Q}\right)} \lesssim \varepsilon.$$

**Putting things together.** Using the data processing inequality, we have

$$\mathsf{TV}(\overline{Q}, \widehat{Q}) \le \mathsf{TV}(q_T, \mathcal{N}(0, I_d)).$$

Combining these results, we have

$$\mathsf{TV}(Q, \widehat{Q}) \le \mathsf{TV}(Q, Q_{\text{dis}}) + \mathsf{TV}(Q_{\text{dis}}, \widetilde{Q}) + \mathsf{TV}(\widetilde{Q}, \overline{Q}) + \mathsf{TV}(\overline{Q}, \widehat{Q})$$
$$\lesssim \delta + \varepsilon + \mathsf{TV}(Q, Q_{\text{dis}}) + \mathsf{TV}(q_T, \mathcal{N}(0, I_d)).$$

Since $q_\gamma$ is the distribution for $x_{T-t_N}$ under $Q$ and $p_{T-t_N}$ is the distribution for $x_{T-t_N}$ under $\widehat{Q}$, we have

$$\mathsf{TV}(q_\gamma, p_{T-t_N}) \le \mathsf{TV}(Q, \widehat{Q}) \lesssim \delta + \varepsilon + \mathsf{TV}(Q, Q_{\text{dis}}) + \mathsf{TV}(q_T, \mathcal{N}(0, I_d)).$$

$\square$

**Lemma 3.1** (Main Sampling Lemma). *Consider an arbitrary sequence of discretization times* $0 = t_0 < t_1 < \cdots < t_N = T - \gamma$. *Assume that for each* $k \in \{0, \dots, N-1\}$, *the following holds:*

$$D_{q_{T-t_k}}^{\delta/N}(\widehat{s}_{T-t_k}, s_{T-t_k}) \le \frac{\varepsilon}{\sigma_{T-t_k}} \cdot \frac{1}{\sqrt{T + \log \frac{1}{\gamma}}}$$

*Then, the output distribution* $\widehat{q}_{T-t_N}$ *satisfies*

$$\mathsf{TV}(\widehat{q}_{T-t_N}, q_{T-t_N}) \lesssim \delta + \varepsilon + \mathsf{TV}(Q, Q_{dis}) + \mathsf{TV}(q_T, \mathcal{N}(0, I_d))$$

*Proof.* Follows by Lemma B.1 and Lemma B.2. $\square$

The next two lemmas from existing works show that the discretization error, $\mathsf{TV}(Q, Q_{\text{dis}})$, is relatively small. Furthermore, as $T$ increases, $q_T$ converges exponentially towards $\mathcal{N}(0, I_d)$.

**Lemma B.3** (Discretization Error, Corollary 1 and eq. (17) in Benton et al. (2023)). *For any* $T \ge 1$, $\gamma < 1$ *and* $N \ge \log(1/\gamma)$, *there exists a sequence of* $N$ *discretization times such that*

$$\mathsf{TV}(Q, Q_{dis}) \lesssim \sqrt{\frac{d}{N}} \left( T + \log \frac{1}{\gamma} \right).$$

**Lemma B.4** (TV between true Gaussian and $q_T$ for large $T$, Proposition 4 in Benton et al. (2023)). *Let $q$ be a distribution with a finite second moment of* $m_2^2$. *Then, for* $T \ge 1$ *we have*

$$\mathsf{TV}(q_T, \mathcal{N}(0, I_d)) \lesssim (\sqrt{d} + m_2)e^{-T}.$$

Combining lemma B.3 and lemma B.4 with lemma 3.1, we have the following result:

**Corollary B.5.** *Let $q$ be a distribution with finite second moment* $m_2^2$. *For any* $T \ge 1$, $\gamma < 1$ *and* $N \ge \log(1/\gamma)$, *there exists a sequence of discretization times* $0 = t_0 < t_1 < \cdots < t_N = T - \gamma$ *such that if the following holds for each* $k \in \{0, \dots, N-1\}$:

$$D_{q_{T-t_k}}^{\delta/N}(\widehat{s}_{T-t_k}, s_{T-t_k}) \le \frac{\varepsilon}{\sigma_{T-t_k}},$$

*then there exists a sequence of* $N$ *discretization times such that*

$$\mathsf{TV}(q_\gamma, p_{T-t_N}) \lesssim \delta + \varepsilon \sqrt{T + \log \frac{1}{\gamma}} + \sqrt{\frac{d}{N}} \left( T + \log \frac{1}{\gamma} \right) + (\sqrt{d} + m_2)e^{-T}.$$

This implies our main theorem of this section as a corollary.

**Theorem B.6.** *Let $q$ be a distribution with finite second moment* $m_2^2$. *For any* $\gamma > 0$, *there exist* $N = \widetilde{O}(\frac{d}{\varepsilon^2 + \delta^2} \log^2 \frac{d + m_2}{\gamma})$ *discretization times* $0 = t_0 < t_1 < \cdots < t_N < T$ *such that if the following holds for every* $k \in \{0, \dots, N-1\}$,

$$D_{q_{T-t_k}}^{\delta/N}(\widehat{s}_{T-t_k}, s_{T-t_k}) \le \frac{\varepsilon}{\sigma_{T-t_k}}$$

*then the SDE process in equation 5 can produce a sample from a distribution that is within* $\widetilde{O}(\delta + \varepsilon \sqrt{\log((d + m_2)/\gamma)})$ *in* $\mathsf{TV}$ *distance of* $q_\gamma$ *in* $N$ *steps.*

*Proof.* By setting $T = \log(\frac{\sqrt{d}+m_2}{\varepsilon+\delta})$ and $N = \frac{d(T+\log(1/\gamma))^2}{\varepsilon^2+\delta^2}$ in corollary B.5, we have

$$\mathsf{TV}(q_\gamma, p_{T-t_N}) = \widetilde{O}\left(\delta + \varepsilon\sqrt{\log\frac{d+m_2}{\gamma}}\right).$$

$\square$

Furthermore, we present our theorem under the case when $m_2$ lies between $1/\text{poly}(d)$ and $poly(d)$ to provide a clearer illustration.

**Theorem 1.2.** *Let $q$ be a distribution over $\mathbb{R}^d$ with second moment $m_2^2$ between $1/poly(d)$ and $poly(d)$. For any $\gamma > 0$, there exist $N = \widetilde{O}(\frac{d}{\varepsilon^2+\delta^2}\log^2\frac{1}{\gamma})$ discretization times $0 = t_0 < t_1 < \cdots < t_N < T$ such that if the following holds for every $k \in \{0, \ldots, N-1\}$:*

$$D_{q_{T-t_k}}^{\delta/N}(\widehat{s}_{T-t_k}, s_{T-t_k}) \leq \frac{\varepsilon}{\sigma_{T-t_k}}$$

*then the SDE process in equation 5 can produce a sample from a distribution that is within $\widetilde{O}(\delta + \varepsilon\sqrt{\log(d/\gamma)})$ in $\mathsf{TV}$ distance to $q_\gamma$ in $N$ steps.*

## C  END-TO-END GUARANTEE

In this section, we state our end-to-end guarantee that combines our score estimation result in the new equation 8 sense with prior sampling results (from Benton et al. (2023)) to show that the score can be learned using a number of samples scaling polylogarithmically in $\frac{1}{\gamma}$, where $\gamma$ is the desired sampling accuracy.

**Corollary C.1.** *Let $q$ be a distribution of $\mathbb{R}^d$ with second moment $m_2^2$. For any $\gamma > 0$, there exist $N = \widetilde{O}(\frac{d}{\varepsilon^2+\delta^2}\log^2\frac{m_2+1/m_2}{\gamma})$ discretization times $0 = t_0 < \cdots < t_N < T$ such that if $\mathcal{H}$ contain approximations $h_{T-t_k}$ that satisfy*

$$\mathbb{E}_{x \sim q_t}\left[\|h_{T-t_k}(x) - s_{T-t_k}(x)\|^2\right] \leq \frac{\delta \cdot \varepsilon^3}{CN^2\sigma_{T-t_k}^2} \cdot \frac{1}{\log\frac{d+m_2+1/m_2}{\gamma}}$$

*for sufficiently large constant $C$, then given $m = \widetilde{O}\left(\frac{N}{\varepsilon^3}(d + \log\frac{1}{\delta})\log\frac{|\mathcal{H}|}{\delta}\log\frac{m_2+1/m_2}{\gamma}\right)$ samples, with $1 - \delta$ probability the SDE process in equation 5 can sample from a distribution $\varepsilon$-close in $\mathsf{TV}$ to a distribution $\gamma m_2$-close in 2-Wasserstein to $q$ in $N$ steps.*

*Proof.* Note that for an arbitrary $t > 0$, the 2-Wasserstein distance between $q$ and $q_t$ is bounded by $O(tm_2 + \sqrt{t}d)$. Therefore, by choosing $t_N = T - \min(\gamma, \gamma^2 m_2^2/d)$, Theorem B.6 shows that by choosing $N = \widetilde{O}(\frac{d}{\varepsilon'^2+\delta^2}\log^2\frac{d+m_2}{\min(\gamma,\gamma^2 m_2^2/d)})$, we only need

$$D_{q_{T-t_k}}^{\varepsilon/N}(\widehat{s}_{T-t_k}, s_{T-t_k}) \leq \frac{\varepsilon'}{\sigma_{T-t_k}\sqrt{\log\frac{d+m_2}{\min(\gamma,\gamma^2 m_2^2/d)}}}$$

then DDPM can produce a sample from a distribution within $\widetilde{O}(\varepsilon')$ in $\mathsf{TV}$ distance to a distribution $\gamma m_2$-close in 2-Wasserstein to $q$ in $N$ steps. Note that

$$\log\frac{d+m_2}{\min(\gamma,\gamma^2 m_2^2/d)} \lesssim \log\frac{d+m_2+1/m_2}{\gamma}.$$

Therefore, we only need to take $\widetilde{O}(\frac{d}{\varepsilon'^2+\delta^2}\log^2\frac{m_2+1/m_2}{\gamma})$ steps. Therefore, to achieve this, we set $\delta_{\text{train}} = \delta$, $\delta_{\text{score}} = \varepsilon'$, and $\varepsilon = \varepsilon'/\sqrt{\log\frac{d+m_2+1/m_2}{\gamma}} \lesssim \varepsilon'/\sqrt{\log\frac{d+m_2}{\min(\gamma,\gamma^2 m_2^2/d)}}$ in theorem A.3. This gives us the result that with

$$m = \widetilde{O}\left(\frac{N}{\varepsilon'^3}(d + \log\frac{1}{\delta})\log\frac{|\mathcal{H}|}{\delta}\log\frac{m_2+1/m_2}{\gamma}\right)$$

samples, we can satisfy the score requirement given the assumption in the statement. $\square$

Again, we present this corollary under the case when $m_2$ lies between $1/\text{poly}(d)$.

**Corollary 1.3** (End-to-end Guarantee). *Let $q$ be a distribution over $\mathbb{R}^d$ with second moment $m_2^2$ between $1/poly(d)$ and $poly(d)$. For any $\gamma > 0$, there exist $N = \widetilde{O}(\frac{d}{\varepsilon^2+\delta^2}\log^2\frac{1}{\gamma})$ discretization times $0 = t_0 < \cdots < t_N < T$ such that if $\mathcal{H}$ contain approximations $h_{T-t_k}$ that satisfy*

$$\mathop{\mathbb{E}}_{x\sim q_t}\left[\|h_{T-t_k}(x) - s_{T-t_k}(x)\|^2\right] \leq \frac{\delta \cdot \varepsilon^3}{CN^2\sigma_{T-t_k}^2} \cdot \frac{1}{\log\frac{d}{\gamma}}$$

*for sufficiently large constant $C$, then given $m = \widetilde{O}\left(\frac{N}{\varepsilon^3}(d + \log\frac{1}{\delta})\log\frac{|\mathcal{H}|}{\delta}\log\frac{1}{\gamma}\right)$ samples, with $1 - \delta$ probability the SDE process in equation 5 can sample from a distribution $\varepsilon$-close in TV to a distribution $\gamma m_2$-close in 2-Wasserstein to $q$ in $N$ steps.*

## D    HARDNESS OF LEARNING IN $L^2$

In this section, we give proofs of the hardness of the examples we mention in Section 4.

**Lemma 4.1.** *Let $R$ be sufficiently large. Let $p_1$ be the distribution $(1-\eta)\mathcal{N}(0,1) + \eta\mathcal{N}(-R,1)$ with corresponding score function $s_1$, and let $p_2$ be $(1-\eta)\mathcal{N}(0,1) + \eta\mathcal{N}(R,1)$ with score $s_2$. Then, given $m$ samples from either distribution, it is impossible to distinguish between $p_1$ and $p_2$ for $\eta < \frac{1}{m^{1.1}}$ with probability larger than $1/2 + o_m(1)$. But,*

$$\mathop{\mathbb{E}}_{x\sim p_1}\left[\|s_1(x) - s_2(x)\|^2\right] \gtrsim \eta R^2 \quad and \quad \mathop{\mathbb{E}}_{x\sim p_2}\left[\|s_1(x) - s_2(x)\|^2\right] \gtrsim \eta R^2.$$

*Proof.*

$$\mathsf{TV}(p_1, p_2) \gtrsim \eta$$

So, it is impossible to distinguish between $p_1$ and $p_2$ with fewer than $O\left(\frac{1}{\eta}\right)$ samples with probability $1/2 + o_m(1)$.

The score $L^2$ bound follows from calculation. $\qquad\square$

**Lemma 4.2.** *Let $S$ be sufficiently large. Consider the distribution $\widehat{p} = \eta\mathcal{N}(0,1) + (1-\eta)\mathcal{N}(S,1)$ for $\eta = \frac{Se^{-\frac{S^2}{2}+10\sqrt{\log m}\cdot S}}{10\sqrt{\log m}}$, and let $\widehat{s}$ be its score function. Given $m$ samples from the standard Gaussian $p^* = \mathcal{N}(0,1)$ with score function $s^*$, with probability at least $1 - \frac{1}{poly(m)}$,*

$$\widehat{\mathbb{E}}\left[\|\widehat{s}(x) - s^*(x)\|^2\right] \leq e^{-O(S\sqrt{\log m})} \quad but \quad \mathop{\mathbb{E}}_{x\sim p^*}\left[\|\widehat{s}(x) - s^*(x)\|^2\right] \gtrsim \frac{S^2}{m}.$$

*Proof.* Let $X_1, \ldots, X_m \sim p^*$ be the $m$ samples from $\mathcal{N}(0,1)$. With probability at least $1 - \frac{1}{poly(m)}$, every $X_i \leq 2\sqrt{\log m}$. Now, the score function of the mixture $\widehat{p}$ is given by

$$\widehat{s}(x) = \frac{-x - (x - S)\left(\frac{1-\eta}{\eta}\right)e^{-\frac{S^2}{2}+Sx}}{1 + \left(\frac{1-\eta}{\eta}\right)e^{-\frac{S^2}{2}+Sx}}$$

For $x \leq 2\sqrt{\log m}$,

$$\widehat{s}(x) = -x\left(1 + \frac{e^{-O(S\sqrt{\log m})}}{S}\right) + e^{-O(S\sqrt{\log m})}$$

So,

$$\widehat{\mathbb{E}}\left[\|\widehat{s}(x) - s^*(x)\|^2\right] \leq e^{-O(S\sqrt{\log m})}$$

On the other hand,

$$\mathop{\mathbb{E}}_{x\sim p^*}\left[\|\widehat{s}(x) - s^*(x)\|^2\right] \gtrsim \frac{S^2}{m}$$

$\qquad\square$

# E  UTILITY RESULTS

**Lemma E.1** (From Gupta et al. (2023)). *Let $f$ be an arbitrary distribution on $\mathbb{R}^d$, and let $f_\Sigma$ be the $\Sigma$-smoothed version of $f$. That is, $f_\Sigma(x) = \mathbb{E}_{y \sim f}\left[(2\pi)^{-d/2} \det(\Sigma)^{-1/2} \exp\left(-\frac{1}{2}(x-Y)^T \Sigma^{-1}(x-Y)\right)\right]$. Let $s_\Sigma$ be the score function of $f_\Sigma$. Let $(X, Y, Z_\Sigma)$ be the joint distribution such that $Y \sim f$, $Z_\Sigma \sim \mathcal{N}(0, \Sigma)$ are independent, and $X = Y + Z_\Sigma \sim f_\Sigma$. We have for $\varepsilon \in \mathbb{R}^d$,*

$$\frac{f_\Sigma(x + \varepsilon)}{f_\Sigma(x)} = \mathop{\mathbb{E}}_{Z_\Sigma | x} \left[ e^{-\varepsilon^T \Sigma^{-1} Z_\Sigma - \frac{1}{2} \varepsilon^T \Sigma^{-1} \varepsilon} \right]$$

*so that*

$$s_\Sigma(x) = \mathop{\mathbb{E}}_{Z_\Sigma | x} \left[ -\Sigma^{-1} Z_\Sigma \right]$$

**Lemma E.2** (From Hsu et al. (2012), restated). *Let $x$ be a mean-zero random vector in $\mathbb{R}^d$ that is $\Sigma$-subgaussian. That is, for every vector $v$,*

$$\mathbb{E}\left[ e^{\lambda \langle x, v \rangle} \right] \le e^{\lambda^2 v^T \Sigma v / 2}$$

*Then, with probability $1 - \delta$,*

$$\|x\| \lesssim \sqrt{\mathrm{Tr}(\Sigma)} + \sqrt{2\|\Sigma\| \log \frac{1}{\delta}}$$

**Lemma E.3.** *For $s_\Sigma$ the score function of an $\Sigma$-smoothed distribution where $\Sigma = \sigma^2 I$, we have that $v^T s_\Sigma(x)$ is $O(1/\sigma^2)$-subgaussian, when $x \sim f_\Sigma$ and $\|v\| = 1$.*

*Proof.* We have by Lemma E.1 that

$$s_\Sigma(x) = \mathop{\mathbb{E}}_{Z_\Sigma | x} \left[ \Sigma^{-1} Z_\Sigma \right]$$

So,

$$
\begin{aligned}
\mathop{\mathbb{E}}_{x \sim f_\Sigma} \left[ (v^T s_\Sigma(x))^k \right] &= \mathop{\mathbb{E}}_{x \sim f_\Sigma} \left[ \mathop{\mathbb{E}}_{Z_\Sigma | x} \left[ v^T \Sigma^{-1} Z_\Sigma \right]^k \right] \\
&\le \mathop{\mathbb{E}}_{Z_\Sigma} \left[ (v^T \Sigma^{-1} Z_\Sigma)^k \right] \\
&\le \frac{k^{k/2}}{\sigma^k} \quad \text{since } v^T Z_\Sigma \sim \mathcal{N}(0, \sigma^2)
\end{aligned}
$$

The claim follows. $\square$

**Lemma E.4.** *Let $\Sigma = \sigma^2 I$, and let $x \sim f_\Sigma$. We have that with probability $1 - \delta$,*

$$\|s_\Sigma(x)\|^2 \lesssim \frac{d + \log \frac{1}{\delta}}{\sigma^2}$$

*Proof.* Follows from Lemmas E.3 and E.2. $\square$

**Lemma E.5.** *For $z \sim \mathcal{N}(0, \sigma^2 I_d)$, with probability $1 - \delta$,*

$$\left\| \frac{z}{\sigma^2} \right\| \lesssim \frac{\sqrt{d + \log \frac{1}{\delta}}}{\sigma}$$

*Proof.* Note that $\|z\|^2$ is chi-square, so that we have for any $0 \le \lambda < \sigma^2/2$,

$$\mathop{\mathbb{E}}_{z} \left[ e^{\lambda \| \frac{z}{\sigma^2} \|^2} \right] \le \frac{1}{(1 - 2(\lambda/\sigma^2))^{d/2}}$$

The claim then follows by the Chernoff bound. $\square$

**Lemma E.6.** *For $x \sim f_\Sigma$, with probability $1 - \delta$,*

$$\mathbb{E}_{Z_\Sigma | x} \left[ \frac{\|Z_\Sigma\|}{\sigma^2} \right] \lesssim \frac{\sqrt{d + \log \frac{1}{\delta}}}{\sigma}$$

*Proof.* Since $Z_\Sigma \sim \mathcal{N}(0, \sigma^2 I_d)$ so that $\|Z_\Sigma\|^2$ is chi-square, we have that for any $0 \leq \lambda < \sigma^2/2$, by Jensen's inequality,

$$\mathbb{E}_{x \sim f_\Sigma} \left[ e^{\lambda \mathbb{E}_{Z_\Sigma | x} \left[ \frac{\|Z_\Sigma\|}{\sigma^2} \right]^2} \right] \leq \mathbb{E}_{Z_\Sigma} \left[ e^{\lambda \frac{\|Z_\Sigma\|^2}{\sigma^4}} \right] \leq \frac{1}{(1 - 2(\lambda/\sigma^2))^{d/2}}$$

The claim then follows by the Chernoff bound. That is, setting $\lambda = \sigma^2/4$, for any $t > 0$,

$$\Pr_{x \sim f_\Sigma} \left[ \mathbb{E}_{Z_\Sigma | x} \left[ \frac{\|Z_\Sigma\|}{\sigma^2} \right]^2 \geq t \right] \leq \frac{\mathbb{E}_{x \sim f_\Sigma} \left[ e^{\lambda \mathbb{E}_{Z_\Sigma | x} \left[ \frac{\|Z_\Sigma\|}{\sigma^2} \right]^2} \right]}{e^{\lambda t}} \leq 2^{d/2} e^{-t\sigma^2/4} = 2^{\frac{d \ln 2}{2} - \frac{t\sigma^2}{4}}$$

For $t = O\left( \frac{d + \log \frac{1}{\delta}}{\sigma^2} \right)$, this is less than $\delta$. $\qquad\square$

**Theorem E.7** (Girsanov's theorem). *For $t \in [0, T]$, let $\mathcal{L}_t = \int_0^t b_s \, \mathrm{d}B_s$ where $B$ is a $Q$-Brownian motion. Assume Novikov's condition is satisfied:*

$$\mathbb{E}_Q \left[ \exp \left( \frac{1}{2} \int_0^T \|b_s\|_2^2 \, \mathrm{d}s \right) \right] < \infty.$$

*Then*

$$\mathcal{E}(\mathcal{L})_t := \exp \left( \int_0^t b_s \, \mathrm{d}B_s - \frac{1}{2} \int_0^t \|b_s\|_2^2 \, \mathrm{d}s \right)$$

*is a $Q$-martingale and*

$$\widetilde{B}_t := B_t - \int_0^t b_s \, \mathrm{d}s$$

*is a Brownian motion under $P$ where $P := \mathcal{E}(\mathcal{L})_T Q$, the probability distribution with density $\mathcal{E}(\mathcal{L})_T$ w.r.t. $Q$.*

