# OpenReview forum: "Sample-Efficient Training for Score-Based Diffusion"
_ICLR.cc/2024/Conference — Submitted to ICLR 2024_

### Official Review · Reviewer_yv9g · 2023-10-30

**Soundness:** 3 good
**Presentation:** 2 fair
**Contribution:** 3 good
**Rating:** 6
**Confidence:** 3

**Summary:**

The paper addresses the sample complexity associated with training score-based diffusion models. It introduces an essential concept: the $1-\delta$ error, a more robust measure compared to the conventional L2 error metric. It is demonstrated that using the $1-\delta$ error, efficient training can be achieved with a sample complexity of poly($log (m^2/\gamma)$) when employing score matching.This new measure proves sufficient for enabling efficient sampling through reverse SDE.

**Strengths:**

1. The paper introduces the concept of the $1-\delta$ error as a more robust measure for diffusion models, which is a novel and innovative contribution to the field. This new measure offers an alternative approach to assessing score estimation precision;

2. The paper effectively addresses the crucial issue of sample complexity in training diffusion models. It recognizes the challenge of balancing score estimation accuracy with the number of required samples, providing insights into how to achieve efficient training.

3. By introducing the $1-\delta$ error as a foundational measure for future work, the paper provides a valuable reference point for researchers looking to advance the field of diffusion models.

**Weaknesses:**

I believe the presentation of this paper can be improved and there are several typos:
1. After equation (1) on page 1, it writes 'for Brownian motion $dB_t$', while $dB_t$ is NOT the Brownian motion (actually $B_t$ is);
2. On page 1, it writes $x_t\sim e^{-t} x_0 + N(0, \sigma_t^2), which seems like $x_t$ is restricted in $\mathbb{R}$, but after equation (2) it looks like $x_t$ is a process in $\mathbb{R}^d$ before $d$ is even defined;
3. Also on page 1, it writes 'logarithmic in $m_2/\epsilon$', where neither $m_2$ or $\epsilon$ is defined before. Actually $m_2$ is defined in Theorem 1.2 for the first time;
4. Equation (4) seems to refer to the minimizer, then it should be 'argmin ...' instead of 'min ...';
5. Before equation (5), I can understand what you mean by 'with norm bounded by R', but I think it's not a commonly used expression. Maybe you can try something like 'supported in B(0,R)' instead?
6. Before equation (6), we see $m_2$ again, still undefined;
7. After equation (6), 'to satisfy' should be 'to be satisfied' instead;
8. In Lemma 4.1, the parameter $\eta$ depends on $m$, while $m$ is defined in the proof;
9. In Lemma 4.1, I guess $\hat{\mathbb{E}}$ means the empirical expectation, but it is not defined in the paper.
10. In Theorem A.1, the notation of $s_r$, $s^*_r$, $\hat{s}_r$ and $\tilde{s}_r$ is a bit messy.
11. A typo! The title of Appendix D should be 'Utility Results' instead of 'Utility Resuts' I think.

**Questions:**

1. In the part named Our results, what's the intuition of 'one would like to show ... with poly($log \frac{m_2}{\gamma}$)'? Where does this poly($log \frac{m_2}{\gamma}$) come from?
2. In the remark after Corollary 1.1, it states that 'Corollary 1.1 doesn’t depend on the domain size at all'. But H contains sufficiently accurate approximations to each score. Does it mean that $|H|$ should be related to the domain size?
3. How is the discussion in section 4 related to that in Appendix D of [1]?
4. On page 5, it states that 'If x is bounded by some value $R$, then the score would be bounded by $R/\sigma^2$. Can you prove it or provide a reference?
5. In Theorem A.1, what does it mean by 'r-smoothed version'?
6. In Theorem A.1, are $\hat{s}_r$ and $\tilde{s}_r$ different? It seems that in the proof you assume they are equal (but actually they are not I think).

[1] Holden Lee, Jianfeng Lu, and Yixin Tan. Convergence for score-based generative modeling with polynomial complexity. arXiv preprint arXiv:2206.06227, 2022.

---

> ### Author Response · Authors · 2023-11-21
>
> We appreciate your positive reception of the concept of the $1 - \delta$ error we introduced. As you suggested, we have uploaded a revised manuscript with a clearer presentation.
> In particular, we appreciate and have addressed your corrections and suggestions such as defining $m_2$ before using it, and correcting several typos.
>
> **Explanation of $\text{poly}(\log(m_2/\gamma))$:** This is motivated by the discussion in the previous section, where we note that prior works improve a $\text{poly}(m_2/\gamma)$ to a $\text{poly}(\log(m_2/\gamma))$ for *sampling*. The natural question is to see if this improvement is possible in *training*, which is what we investigate.
>
> **Domain size:**
> You make a fair point, $\mathcal{H}$ doesn't necessarily depend on the domain size but it typically will.  For neural networks, for example, this induces a logarithmic dependence on the domain size. (See Theorem 1.5).
>
>
> **Appendix D of Lee et al:** The two examples are essentially the opposite of each other. They give an example of distributions with small $L^2$ error and large TV error; we give distributions with large $L^2$ error and small TV error.
> They show non-annealed Langevin dynamics fails even with small $L^2$ error; we show annealed Langevin dynamics succeeds even for large $L^2$ error.
>
> Note also that the conclusion of Lee et al explicitly asks the question we address in this work:
>
> > The assumption that we have a score estimate that is
> $O(1)$-accurate in $L^2$, although weaker than the usual assumptions for theoretical analysis, is in fact still a
> strong condition in practice that seems unlikely to be satisfied (and difficult to check) when learning complex
> distributions such as distributions of images. What would a reasonable weaker condition be, and in what
> sense can we still obtain reasonable samples?
>
> Our distance measure $D_p^\delta$ is a weaker condition than $L^2$ that is easier to learn in but still yields reasonable samples.
>
> **Bound on score:**
> This is implied by Tweedie's formula: $s(x) = \mathbb{E}_{z|x}[-z/\sigma^2]$, and the numerator is $O(R)$.
>
> **Difference between $\hat{s}$ and $\tilde{s}$:** $\tilde{s}$ refers to some candidate score function in $\mathcal{H}$ that we are trying to analyze the loss function evaluated for, while $\hat{s}$ is the empirical loss minimizer among these.

---

> > ### Comment · Reviewer_yv9g · 2023-12-01
> >
> > My questions are mostly addressed and hence I raised my score.

---

### Official Review · Reviewer_KNkT · 2023-11-01

**Soundness:** 2 fair
**Presentation:** 3 good
**Contribution:** 2 fair
**Rating:** 5
**Confidence:** 3

**Summary:**

This paper shows that estimating the score in L2 requires this polynomial dependence, but polylogarithmic samples actually do suffice for sampling.

**Strengths:**

The investigation of sample complexity is a core issue in statistics literature. For modern generative models, it is quite interesting to see the sample dependency of these systems.

This paper find a good angle that it is not necessary to learn the score accurately
Which is particularly challenging. This idea can help to justify the success of current diffusion models

**Weaknesses:**

The writing of the paper should be improved. It contains too much previous work and technical details. The contribution of this work are scattered.

Too many informal results, which makes readers hard to determine which parts are not rigorous.

The setting looks artificial compared to true score-based model.

**Questions:**

Take Figure 1 for example, in a real score-based model, we would not evaluate score near 0, because we construct the whole process from p to N(0,I). Then we only evaluate the score p_T near 0 rather than p_0. Since the p_T is highly smoothed, and the importance can also be reflected by the number of samples, the score is generally well-estimated in L2 sense. Otherwise the mixing of the algorithm is incorrect. In [1], it is shown that the score-based algorithm has a good mixing property. From this view, the proposed metric looks a bug, not feature?
Can you explain this?

I understand the challenges and hardness of learning L2, but there are still more examples needed to justify the new metric.

---

> ### Author Response · Authors · 2023-11-21
>
> Thank you for your review. We have made several changes to the manuscript with your comments in mind. In particular, we have added formal statements to the main text to make it clearer what our results are.
>
> We clarify some questions you brought up below:
>
> **The setting looks artificial compared to true score-based model.**
>
> We have no idea what you mean here.  Our setting is exactly that of DDPM, as deployed in settings like Stable Diffusion.  You train a neural network with score matching; you sample by discretizing the SDE.  The question is: how many training samples do you need for accurate sampling?
>
> **Example.**
> You are correct that on this example, the actual sampling procedure will have no problem.  But the *existing theory* does: the score functions are not learned in $L^2$, so the existing theory cannot say the sampling will be accurate.
>
> Your argument is essentially the intuition for our paper: it's OK to have large $L^2$ error on average, as long as the squared error is "generally" small (see our distance measure (8)).

---

### Official Review · Reviewer_uE9j · 2023-11-01

**Soundness:** 3 good
**Presentation:** 3 good
**Contribution:** 2 fair
**Rating:** 6
**Confidence:** 4

**Summary:**

This paper studies the complexity of training and sampling using score-based diffusion models. The authors focus on the sample complexity, i.e., the number of samples needed to reach a given accuracy measured e.g. in TV or Wasserstein-2 distance. The key result is an improvement from ${\rm poly}(R/\gamma)$ to ${\rm poly}\log(m_2/\gamma)$ samples, where $R$ is the bound on the norm of the distribution, $\gamma$ the required accuracy in Wasserstein-2 distance, and $m_2$ the second moment. The scaling with respect to the other parameters (input dimension, 1/accuracy in TV distance) remains polynomial.

The idea is to consider a less restrictive measure for the estimation of the score: instead of the L2 norm, the authors propose a form of quantile error in which regions with low probability are not considered. In fact, they authors show that the estimation of the score in L2 requires polynomially many samples in $1/\gamma$ via an explicit example.

**Strengths:**

* The improvement in the sample complexity in terms of the accuracy in Wasserstein distance is significant: from polynomial to logarithmic.

* The idea of using a different notion of distance to evaluate the error in the score estimation is new and could be useful more broadly when analysing score-based diffusion models.

* I also appreciated the explicit counterexample on the difficulty of learning the score in L2.

**Weaknesses:**

* The scope of the paper is quite limited. While the improvement in terms of accuracy in Wasserstein distance is remarkable, the dependency on the accuracy in TV remains polynomial. What's the point of improving drastically the accuracy in W2, if the accuracy in TV remains bad? For this reason, while I like the idea of using the quantile measure, the benefit of doing so in the context of score-based diffusion remains unclear.

* The complexity is also polynomial in $d$. Recent papers analysing diffusion models (see e.g. Table 1 in "Linear convergence bounds for diffusion models via stochastic localization") show that this dependency is linear or quadratic in $d$. I appreciate that the setting of this paper is different (most works assume access to a good L2 score, which is shown to be impossible if one sticks with a logarithmic dependency in $1/\gamma$). Nonetheless, the authors should track how the bound scales in $d$ and compare to existing work.

* It would also add value to the paper to track the dependency on something more explicit than the cardinality of the hypothesis class $|\mathcal H|$. Along the same lines, for the result on neural networks (Theorem 3.1), one needs to assume the existence of a network that approximates the score well enough.

* The novelty in terms of proof is not high. Basically the idea is to exclude a region with low probability and then use existing analyses (mostly, Benton et al., 2023). This is admittedly a minor point (simple ideas can be very useful!). The two key weaknesses above are the main reason of my score.

**Questions:**

Can the authors comment on the points raised above?

---

> ### Author Response · Authors · 2023-11-21
>
> Thank you for your review. Below we have addressed the issues that you brought up, and hope that this helps clarify and explain the theory.
>
> **Scope:** "While the improvement in terms of accuracy in Wasserstein distance is remarkable, the dependency on the accuracy in TV remains polynomial."
>
> Learning a distribution within TV error $\varepsilon$ requires $1 / \varepsilon^2$ samples information theoretically, so we cannot hope for going beyond polynomial for the dependency on the accuracy in TV.
>
> "What's the point of improving drastically the accuracy in W2, if the accuracy in TV remains bad?"
>
> The W2 and TV error are incommensurate.  The W2 error is "how blurry is the result", while the TV error is "what fraction of the distribution are completely missed".  These are on different scales; if you want very sharp images, you want low W2 error, and we show that this can be achieved.
>
> **The authors should track how the bound scales in d and compare to existing work.**
> The new version presents this.  For *sampling*, our bound is identical to the best existing analysis of the SDE (so, $O(d)$).  For *training*, estimating a single score (Theorem 1.1) takes sample complexity linear in $d$; estimating all the scores gives a $d^2$ dependence for Corollary 1.3.  For comparison, using the Block et al. bound would need $d^3$ for Corollary 1.3.  [They give a bound in terms of the Lipschitzness, which can be $\Theta(d)$ in the regime they consider.]
>
> **It would also add value to the paper to track the dependency on something more explicit than the cardinality of the hypothesis class $\mathcal{H}$.**.
>
> What would you propose?  The cardinality is pretty explicit, and leads to good bounds for neural networks (see Theorem 1.5).  One might try the Rademacher complexity of $\mathcal{H}$, but this doesn't work well: the Rademacher complexity scales with the magnitude of the outputs, which (e.g. in the neural network example) is large and will yield a $\text{poly}(1/\gamma)$ dependence.
>
> **Along the same lines, for the result on neural networks (Theorem 3.1), one needs to assume the existence of a network that approximates the score well enough.**
>
> How could it be otherwise?  If you can't represent the score, you aren't going to sample well with score-based diffusion.
>
> **Proof novelty:**
> The idea of the proof is not actually to exclude a region of low probability.  The Block et al analysis shows for each possible score function $s$ that the score matching objective approximates its $L^2$ error, with enough samples.  The problem is that getting this many samples *requires* $\text{poly}(\frac{1}{\gamma})$ samples, because for some score functions $s$ the value in the score matching objective has very large variance.
>
> The idea in our proof is to essentially argue that those score functions are *very* bad, so it's OK to have a poor approximation to just how bad they are.  We show that the *difference* between the score matching objective for the optimal $s^*$ and any candidate $s$ has variance related to its expectation, such that every sufficiently-far $s$ will be rejected with high probability.
>
> The proof involving excluding regions of low probability is just to bridge the gap between existing theory and our new proposed error metric, which we justify the use of in section 4.  Proving convergence in our new error metric is the main contribution.

---

> > ### Comment · Reviewer_uE9j · 2023-12-04
> > **acknowledging authors' reply and raising score**
> >
> > I have read the authors' reply. I still think that the scope of the paper is rather limited. However, some of my concerns are addressed (in particular, that concerning the dependence of $d$), hence I raise my score to 6.

---

### Official Review · Reviewer_3jfL · 2023-11-01

**Soundness:** 3 good
**Presentation:** 2 fair
**Contribution:** 2 fair
**Rating:** 6
**Confidence:** 3

**Summary:**

This paper studies the sample complexity for estimating the score functions in the diffusion model sampling process. They establish a polynomial sample complexity under a robust measure they proposed in the paper. They also apply their results to the sampling procedure by incorporating the convergence rate of Benton et al 2023. Their technique might be helpful to improve the sample complexity bound for diffusion based algorithms.

**Strengths:**

This paper studies the sample complexity of estimating the score functions in diffusion model. To evaluate the estimation error, they propose a robust measure of distance. For their main result, they show that if the true score function can be approximated well using the function class $H$, and $H$ is finite, then $m = poly(d, 1 / \epsilon, 1 / \delta, N, \log |H| / \delta_{train})$ samples is sufficient for getting $\epsilon$ accuract simultaneously for all score functions. This rate does not require the target distribution is bounded, hence generalizes that of Block et al 2020.

**Weaknesses:**

1. I feel the presentation is not clear enough for readers to fully understand the merit of this paper. For more details see my question list.
2. Only informal theorems are presented in the main text. What prevents the authors from adding formal theorems?
3. Corollary 1.1 does not seem very surprising as $H$ is finite. The authors contain proof in their paper. But maybe it would be helpful to explain intuitively why it is interesting to establish Corollary 1.1, and why it is not a straightforward consequence of the uniform law of large numbers.

**Questions:**

1. I think $m_2$ appears a lot before the authors define it as the second moment.
2. I think the statement "approximates $q_0$ up to $\epsilon$ TV error and $\gamma$ Wasserstein-2 error" in introduction section is not accurate. As far as I am concerned, the right expression is "there exists a distributions $q$, such that $W(q, q_0) \leq \gamma$ and $TV(q, \hat q) \leq \epsilon$, where $\hat q$ is outputted by the diffusion model" (at least this is the case in Benton et al 2023). The authors might want to clearly state that to avoid causing confusion.
3. What does it mean by "$H$ contains sufficiently accurate approximations to each score" in Corollary 1.1?
4. What is the relation between the last sentence on page 3 and Theorem 1.2? Theorem 1.2 seems to be independent of training, why can we see from this theorem that our outlier-robust approximation suffices for sampling?
5. Maybe the authors can comment a little bit on the polynomial dependency? Like what is the order of the polynomial. Block et al 2020 has an explicit polynomial dependency, and it is not clear if the results presented here is indeed better if the form of polynomial is not presented.
6.  It is a little bit restrictive to assume $H$ is finite.

---

> ### Author Response · Authors · 2023-11-21
>
> Thank you for your review. We would like to first clarify that our main contribution is not so much about unbounded distributions, as about higher *accuracy*:  to get accuracy $\gamma m_2$ in Wasserstein, our approach proves a $\text{poly}(\log \frac{1}{\gamma})$ sample complexity while prior work needed $\text{poly}(\frac{1}{\gamma})$.  And getting this improved dependence *requires* switching from $L^2$ approximation to a different measure.
>
>
>
> Addressing weaknesses (1, 2): We have rewritten the paper significantly, and included the formal theorem statements in the introduction.
>
> Addressing weakness (3):  First of all, our bound for finite $\mathcal{H}$ is strong enough to extend to continuous neural networks as a corollary by essentially rounding the weights to finite precision (see Theorem 1.5).  So " finite" vs "continuous" is not what makes the result challenging.
>
> Generally speaking, convergence of the empirical mean of a distribution depends on the variance of that distribution.  The score matching objective involves terms involving the norm of the output of a neural network; for the wrong-weight networks, this norm can be *huge*, making the convergence very slow; and to get accuracy $\gamma$, such an approach gives a $\text{poly}(\frac{1}{\gamma})$ rather than  the $\text{poly}(\log \frac{1}{\gamma})$ that we are able to achieve.
>
> Answering specific questions:
>
> 1, 2, 3. Thank you, we have rewritten the introduction more clearly.
>
> 4. Again, we have written this more clearly; sampling requires sufficiently accurate scores, which Theorem 1.1 provides.
>
> 5.  We now include the dependence. It is linear in $d$ and the complexity class $\log \mathcal{H}$. For neural networks (Theorem 1.5), the dependency is linear in $d$, the number of parameters $P$, and the neural network depth $D$.

---

### Author Response · Authors · 2023-11-21
**General Response**

We thank all the reviewers for their helpful comments.
The general theme was that the reviewers found our submission to be "novel and innovative", yet also rightly pointed out that "the presentation is not clear enough".

We have uploaded a significantly revised version of our paper, which we believe is much clearer, and hope addresses the writing concerns.

---

### Meta-Review · Area_Chair_t6Bq · 2023-12-03

**Metareview:**

The authors improve the sample complexity analysis of score-based diffusion models to be logarithmic in the data radius and desired Wasserstein accuracy (where the end guarantee is being $\epsilon$-close to a distribution that is $\delta$-close in TV distance to the data distribution, the guarantee considered in previous works). They show that existing arguments via L^2 score estimation suffer from this lower bound, and instead argue via L^2 accuracy on a high-probability region. They mainly consider finite hypothesis classes, but apply it to neural networks by a covering argument.
This is a new theoretical contribution; however, reviewers pointed out that the presentation could be improved, and various ways the analysis could have been more careful/comprehensive, for instance: using more general notions of dimension of the hypothesis class (e.g., pseudo- or fat-shattering dimension) and being more precise about the powers in polynomial dependences. This appears to be improved in the revision, but would benefit from more careful review.

**Justification For Why Not Higher Score:**

The paper seemed of limited significance, with lukewarm reviewer reception. Various aspects of the theory could be considered more precisely/comprehensively.

**Justification For Why Not Lower Score:**

N/A

---

### Decision · Program_Chairs · 2024-01-16

Reject